# Scalable Offline Model-Based RL with Action Chunks

**Kwanyoung Park**[1]  **Seohong Park**[1]  **Youngwoon Lee**[2]  **Sergey Levine**[1]

UC Berkeley[1]  Yonsei University[2]

https://kwanyoungpark.github.io/MAC/

## Abstract

In this paper, we study whether model-based reinforcement learning (RL), in particular model-based value expansion, can provide a scalable recipe for tackling complex, long-horizon tasks in offline RL. Model-based value expansion fits an on-policy value function using length-$n$ imaginary rollouts generated by the current policy and a learned dynamics model. While larger $n$ reduces bias in value bootstrapping, it amplifies accumulated model errors over long horizons, degrading future predictions. We address this trade-off with an *action-chunk* model that predicts a future state from a sequence of actions (an "action chunk") instead of a single action, which reduces compounding errors. In addition, instead of directly training a policy to maximize rewards, we employ rejection sampling from an expressive behavioral action-chunk policy, which prevents model exploitation from out-of-distribution actions. We call this recipe **Model-Based RL with Action Chunks (MAC)**. Through experiments on highly challenging tasks with large-scale datasets of up to 100M transitions, we show that MAC achieves the best performance among offline model-based RL algorithms, especially on challenging long-horizon tasks.

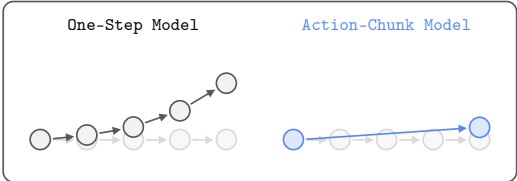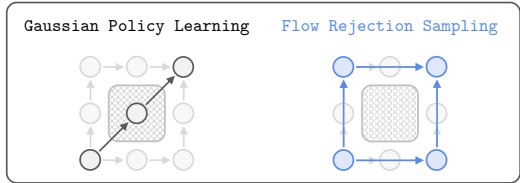

Figure 1: **Two main components of MAC.** (*Left*) Action-chunk models predict a future state given a *sequence* of actions (an "action chunk"), reducing compounding errors and enabling long-horizon model rollouts. (*Right*) Rejection sampling from an expressive (flow) behavioral action-chunk policy enables modeling multi-modal action distributions, while preventing model exploitation from out-of-distribution actions.

## 1 Introduction

Offline reinforcement learning (RL) holds the promise of training effective decision-making agents from data, leveraging large-scale datasets. While offline RL has achieved successes in diverse domains (Kumar et al., 2023; Springenberg et al., 2024), its ability to handle complex, long-horizon tasks remains an open question. Prior work has shown that standard, *model-free* offline RL often struggles to scale to such tasks (Park et al., 2025b), hypothesizing that the cause lies in the pathologies of off-policy, temporal difference (TD) value learning.

In this work, we investigate whether an alternative paradigm, namely *model-based* RL, and in particular model-based value expansion (Feinberg et al., 2018), provides a more effective recipe for long-horizon offline RL. In this recipe, we first train a dynamics model, and fit an *on-policy* value function by rolling out the current policy within the learned model, which is then used to update the policy. Since on-policy value learning has demonstrated promising scalability to long-horizon tasks (Berner et al., 2019; Guo et al., 2025), in contrast to the relatively limited evidence for off-

policy TD learning (Park et al., 2025b), we hypothesize that the combination of on-policy value learning and dynamics modeling may also exhibit strong horizon scalability.

However, there is a tricky trade-off in this recipe. In model-based value expansion, we typically train a value function by rolling out the policy for $n$ steps within the model and regressing toward the target: $V(s_t) \leftarrow \sum_{i=0}^{n-1} \gamma^i r_{t+i} + \gamma^n \bar{V}(s_{t+n})$. Here is the dilemma. On the one hand, we want to use a large $n$ in this value update, as this reduces the *bias* in the bootstrapped target value, $\gamma^n \bar{V}(s_{t+n})$. This is particularly important given that bias accumulation is one of the major factors that hinder the scaling of offline RL (Park et al., 2025b). On the other hand, we want to keep $n$ small enough, as errors in the dynamics model accumulate through autoregressive queries over the horizon. Is there a solution to this trade-off that enables long-horizon model rollouts while preventing error accumulation?

Our main hypothesis in this work is that *action-chunk* models and policies, combined with recent innovations in expressive generative models, can provide a natural solution to the above dilemma, enabling scaling of offline model-based RL to long-horizon tasks. Namely, instead of training a single-step model $p(s_{t+1} \mid s_t, a_t)$, we train a multi-step model $p(s_{t+n} \mid s_t, a_{t:t+n-1})$ that takes an action-chunk $a_{t:t+n-1}$ as input and predicts a future state that is $n$-step ahead. This substantially reduces the number of recursive model calls and mitigates compounding errors (Figure 1, left), enabling long-horizon imaginary rollouts over 100 environment steps.

To use an action-chunk model in the model-based actor-critic framework, we need an action-chunk policy. However, directly training a reward-maximizing action-chunk policy is challenging in offline RL, due to the potentially multi-modal, high-dimensional action-chunk distributions in the dataset (Li et al., 2025). Hence, we employ *rejection sampling* based on samples from an expressive behavioral action-chunk policy trained with flow matching (Lipman et al., 2024). By simply defining the policy as the behavioral action-chunk sample that maximizes the value function, we can not only capture complex action distributions from the dataset (Figure 1, right), but also effectively prevent model exploitation (Kidambi et al., 2020).

We call this recipe **Model-Based RL with Action Chunks (MAC)**, which constitutes the main contribution of this work. Experimentally, we show that MAC vastly improves the horizon scalability of offline model-based RL. In particular, we demonstrate that our scalable model-based RL recipe can consume 100M-scale data to achieve state-of-the-art performance on highly complex, long-horizon robotic manipulation tasks from OGBench (Park et al., 2025a), often outperforming previous model-free and model-based approaches.

## 2 RELATED WORK

**Offline model-free RL.** Offline RL aims to learn a return-maximizing policy from a previously collected dataset, without interaction with the environment (Lange et al., 2012; Levine et al., 2020). As in online RL, offline RL methods can be categorized into model-free and model-based ones. Offline model-free RL methods train a policy without learning a dynamics model. Prior works have proposed a number of model-free approaches based on diverse techniques, such as conservatism (Kumar et al., 2020), behavioral regularization (Wu et al., 2019; Peng et al., 2019; Nair et al., 2020; Fujimoto & Gu, 2021; Tarasov et al., 2023; Park et al., 2025c), uncertainty estimation (An et al., 2021; Nikulin et al., 2023), in-sample maximization (Kostrikov et al., 2022; Xu et al., 2023; Garg et al., 2023), rejection sampling (Chen et al., 2023; Hansen-Estruch et al., 2023), and more (Brandfonbrener et al., 2021; Sikchi et al., 2024).

**Offline model-based RL.** In this work, we focus on offline model-based RL, a paradigm that first trains a dynamics or trajectory model, and then trains a policy based on rollouts generated from the learned model. A line of work trains generative models (*e.g.*, Transformers (Vaswani et al., 2017) and diffusion models (Sohl-Dickstein et al., 2015; Ho et al., 2020)) to model the entire trajectory distribution of the dataset, and typically use conditioning and guidance to compute actions (Chen et al., 2021; Janner et al., 2021b; 2022; Lee et al., 2022; Ajay et al., 2023; Jiang et al., 2023; Li et al., 2023; Chen et al., 2024; Ding et al., 2024; Jackson et al., 2024; Cheng et al., 2025). Another line of work trains a (typically single-step) dynamics model, and trains a policy based on rollouts autoregressively sampled from the learned model. These approaches employ the learned dynamics model for (1) "Dyna"-style data augmentation (Sutton, 1991; Janner et al., 2019; Yu et al., 2020; Kidambi

et al., 2020; Yu et al., 2021; Rigter et al., 2022; Sun et al., 2023; Sims et al., 2024; Lu et al., 2023a), (2) planning (Testud et al., 1978; Argenson & Dulac-Arnold, 2021; Chitnis et al., 2024; Zhou et al., 2025), and (3) value estimation (Feinberg et al., 2018; Jeong et al., 2023; Park & Lee, 2025; Hafner et al., 2025), with diverse techniques to prevent model exploitation and distributional shift, such as ensemble-based uncertainty estimation. Our method is based on model-based value expansion and falls in the third category. However, unlike most of the previous works in this category, we employ an *action-chunk* model instead of a single-step dynamics model to reduce effective horizons and thus error accumulation.

**Horizon reduction and model-based RL.** The curse of horizon is a fundamental challenge in reinforcement learning (Liu et al., 2018; Park et al., 2025b). In the context of model-free RL, previous studies have proposed diverse techniques to reduce effective horizon lengths, such as $n$-step returns to reduce the number of Bellman updates (Sutton & Barto, 2005), and hierarchical policies to reduce the length of the effective policy horizon (Nachum et al., 2018; Park et al., 2023). Long horizons are a central challenge in model-based RL too, since model rollouts suffer from compounding errors as the horizon grows. Prior works in model-based RL address this challenge with trajectory modeling (Janner et al., 2021b; 2022), hierarchical planning (Li et al., 2023; Chen et al., 2024), skill-based action abstraction (Shi et al., 2022), and action-chunk multi-step dynamics modeling (Asadi et al., 2019; Lambert et al., 2021; Zhao et al., 2024; Zhou et al., 2025). Our work is closest to prior works that use action-chunk dynamics models. However, these works either use the action-chunk model only for planning without having the full actor-critic loop (Asadi et al., 2019; Lambert et al., 2021; Zhou et al., 2025), or model the entire state-action chunks (Zhao et al., 2024). Unlike these prior works, we perform *on-policy value learning* with an action-chunk model and policy, while not involving additional planning or full trajectory generation.

## 3 PRELIMINARIES

**Problem setting.** We consider a Markov decision process (MDP) defined as $\mathcal{M} = (\mathcal{S}, \mathcal{A}, r, \mu, p)$, where $\mathcal{S}$ is the state space, $\mathcal{A} = \mathbb{R}^d$ is the action space, $r(s, a) : \mathcal{S} \times \mathcal{A} \to \mathbb{R}$ is the reward function, $\mu(s) \in \Delta(\mathcal{S})$ is the initial state distribution, and $p(s' \mid s, a) : \mathcal{S} \times \mathcal{A} \to \Delta(\mathcal{S})$ is the transition dynamics kernel. $\Delta(\mathcal{X})$ denotes the set of probability distributions on a space $\mathcal{X}$, and we denote placeholder variables in gray. For a policy $\pi(a \mid s) : \mathcal{S} \to \Delta(\mathcal{A})$, we define $V^\pi(s) = \mathbb{E}_{\tau \sim p^\pi(\tau|s_0=s)}[\sum_{t=0}^\infty \gamma^t r(s_t, a_t)]$ and $Q^\pi(s, a) = \mathbb{E}_{\tau \sim p^\pi(\tau|s_0=s, a_0=a)}[\sum_{t=0}^\infty \gamma^t r(s_t, a_t)]$, where $\gamma \in (0, 1)$ denotes the discount factor, $\tau = (s_0, a_0, r_0, s_1, \dots)$ denotes a trajectory, and $p^\pi$ denotes the trajectory distribution induced by $\mu$, $p$, and $\pi$. The goal of offline RL is to find a policy $\pi$ that maximizes $\mathbb{E}_{s_0 \sim \mu(s_0)}[V^\pi(s_0)]$ from an offline dataset $\mathcal{D} = \{\tau^{(i)}\}$ consisting of previously collected trajectories, with no environment interactions.

**Flow matching.** Flow matching (Lipman et al., 2023; Albergo & Vanden-Eijnden, 2023; Liu et al., 2023) is a technique in generative modeling to train a velocity field whose flow generates a target distribution of interest. As with diffusion models (Sohl-Dickstein et al., 2015; Ho et al., 2020), flow models iteratively transform a noise distribution to the target distribution, and have been shown to be highly expressive and scalable (Esser et al., 2024; Lipman et al., 2024).

Formally, assume that we are given a target distribution $p(x) \in \Delta(\mathbb{R}^k)$. For a time-dependent velocity field $v(u, x) : [0, 1] \times \mathbb{R}^k \to \mathbb{R}^k$ (we use $u$ to denote times in flow matching to avoid notational conflicts with environment steps in MDPs), we define its flow, $\psi(u, x) : [0, 1] \times \mathbb{R}^k \to \mathbb{R}^k$, as the unique solution to the following ordinary differential equation (ODE) (Lee, 2012):

$$\frac{\mathrm{d}}{\mathrm{d}u}\psi(u, x) = v(u, \psi(u, x)). \tag{1}$$

Flow matching aims to find a velocity field whose flow transforms a noise distribution (*e.g.*, $k$-dimensional standard Gaussian, $\mathcal{N}(0, I_d)$) at $u = 0$ to the target distribution at $u = 1$.

Prior work (Lipman et al., 2023; Albergo & Vanden-Eijnden, 2023; Liu et al., 2023) has shown that we can train such a velocity field by minimizing the following loss:

$$\mathbb{E}_{\substack{x_0 \sim \mathcal{N}(0, I_d),\ x_1 \sim p(x),\\ u \sim \mathrm{Unif}([0,1]),\ x_u = (1-u)x_0 + ux_1}} \left[ \|v(u, x_u) - (x_1 - x_0)\|_2^2 \right]. \tag{2}$$

We refer to the tutorial by Lipman et al. (2024) for detailed explanations and proofs. After training the velocity field, we can obtain samples from the target distribution by numerically following the velocity field to solve the ODE in practice (*e.g.*, with the Euler method).

## 4 OFFLINE MODEL-BASED RL WITH ACTION CHUNKS

**Motivation.** Our high-level goal is to scale up offline model-based RL to complex, long-horizon decision-making problems. Among model-based RL frameworks, we specifically focus on model-based value expansion (Feinberg et al., 2018), which combines dynamics modeling and *on-policy* value learning. This is because each of these components, namely generative modeling and on-policy RL, has individually been shown to scale to long-horizon tasks (Berner et al., 2019; Harvey et al., 2022; Guo et al., 2025).

In model-based value expansion, we first train a dynamics model, and train an on-policy value function with the following update:

$$V(\hat{s}_t) \leftarrow \sum_{i=0}^{n-1} \gamma^i r(\hat{s}_{t+i}, \hat{a}_{t+i}) + \gamma^n \bar{V}(\hat{s}_{t+n}), \tag{3}$$

where $(s_t = \hat{s}_t, \hat{a}_t, \hat{s}_{t+1}, \ldots, \hat{s}_{t+n})$ is a length-$n$ imaginary rollout sampled from the model using the current policy, and $\bar{V}$ is a target value function. The policy is then updated to maximize the learned value function, and we repeat this procedure.

The problem is: how long should model rollouts be? Unfortunately, we have two seemingly contradictory desiderata.

On the one hand, we want model rollouts to be *long* enough. If $n$ is too small, we end up with a large number of *biased* value updates with short-horizon bootstrapping in Equation (3). This causes the biases to accumulate over the horizon, which is known to be one of the main obstacles hindering value-based RL from scaling to long-horizon tasks (Park et al., 2025b). Hence, we want to keep $n$ large enough.

On the other hand, we want model rollouts to be *short* enough. If we use a standard policy $\pi(a \mid s)$, we need to autoregressively call a learned dynamics model $n$ times to generate a length-$n$ model rollout $(\hat{s}_t, \hat{a}_t, \hat{s}_{t+1}, \ldots, \hat{s}_{t+n})$. This makes errors in the dynamics model accumulate *within* the trajectory chunk, which would degrade performance. Hence, we want to keep $n$ small enough.

Is there a way to naturally resolve this dilemma?

### 4.1 THE IDEA

Our main idea in this work is that a combination of an *action-chunk* policy and an *action-chunk* model can provide a clean solution to the above dilemma, enabling scaling to complex, long-horizon tasks. Specifically, we train an action-chunk model $p(s_{t+n} \mid s_t, a_{t:t+n-1}) : \mathcal{S} \times \mathcal{A}^n \to \Delta(\mathcal{S})$ and an action-chunk policy $\pi(a_{t:t+n-1} \mid s_t) : \mathcal{S} \to \Delta(\mathcal{A}^n)$, where $a_{i:j}$ denotes the action chunk $(a_i, a_{i+1}, \ldots, a_j)$. Since each individual call of the model generates $n$ actions at once, we can reduce the number of recursive model calls by a factor of $n$. This way, we can mitigate both bias accumulation in value learning and error accumulation in model rollouts.

However, several challenges remain in implementing this idea in practice. First, Gaussian policies, used in many previous works in offline model-based RL (Yu et al., 2020; Sun et al., 2023; Lu et al., 2023b; Chitnis et al., 2024; Park & Lee, 2025), are generally not expressive enough to model complex, multi-modal action-*chunk* distributions (Figure 1). Second, penalizing out-of-distribution actions based on uncertainty in the dynamics model, as typically done by prior work in offline model-based RL (Yu et al., 2020; Kidambi et al., 2020; Sun et al., 2023), can be challenging due to the potentially high complexity of the action-chunked dynamics distribution.

To handle these challenges, we employ rejection sampling from an expressive behavioral action-chunk policy. Specifically, we use flow matching (Lipman et al., 2024) to train a behavioral cloning (BC) action-chunk policy, and *define* a policy as the $\arg\max$ action chunk (among $N$ chunks sam-

---

**Algorithm 1** Offline Model-Based RL with Action Chunks (MAC)

---

**Input:** Dataset $\mathcal{D}$, rollout length $H$, action chunking size $n$, rejection sampling size $M$

   *// Training loop*
   **while** not converged **do**
      $\triangleright$ Sample action-chunked batch from the dataset ($\boldsymbol{a}_t = a_{t:t+n-1}$, $\boldsymbol{r}_t = \sum_{i=0}^{n-1} \gamma^i r_{t+i}$)
      Sample batch $\{(s_t, \boldsymbol{a}_t, \boldsymbol{r}_t, s_{t+n})\} \sim \mathcal{D}$

      $\triangleright$ Train **BC policy** using dataset transitions
      Update flow BC policy $\pi_\theta$ with flow-matching loss (Equation (7))
      Update one-step BC policy $\pi_\omega$ with distillation loss (Equation (8))

      $\triangleright$ Train **dynamics** and **reward** model using dataset transitions
      Update dynamics model $p_\psi$ to minimize $\mathbb{E}[\|p_\psi(s_t, \boldsymbol{a}_t) - s_{t+n}\|_2^2]$ (Equation (5))
      Update reward model $r_\psi$ to minimize $\mathbb{E}[\|r_\psi(s_t, \boldsymbol{a}_t) - \boldsymbol{r}_t\|_2^2]$ (Equation (6))

      $\triangleright$ Generate **model rollouts** ($\hat{s}_t = s_t$)
      **for** $k = 0, 1, \ldots, H-1$ **do**
         $\hat{\boldsymbol{a}}_{t+kn} \leftarrow \text{POLICY}(\hat{s}_{t+kn})$
         $\hat{s}_{t+(k+1)n}, \hat{\boldsymbol{r}}_{t+kn} \sim p_\psi(\cdot \mid \hat{s}_{t+kn}, \hat{\boldsymbol{a}}_{t+kn}), r_\psi(\cdot \mid \hat{s}_{t+kn}, \hat{\boldsymbol{a}}_{t+kn})$

      $\triangleright$ Update **value** using model rollouts
      Update value $V_\varphi$ with $nH$-step targets from the rollout (Equation (10))

      $\triangleright$ Learn **critic** for faster rejection sampling
      Update critic $Q_\varphi$ with the learned value function $V_\varphi$ (Equation (11))

   *// Extract action from flow BC policy $\pi_\theta$ with rejection sampling*
   **function** POLICY($s$)
      $z \sim \mathcal{N}(0, I)$
      $\hat{\boldsymbol{a}}^{(i)} = \pi_\omega(s, z)$
      **return** $\text{argmax}_{\hat{\boldsymbol{a}}^{(1)}, \cdots, \hat{\boldsymbol{a}}^{(M)}} Q_\varphi(s, \hat{\boldsymbol{a}}^{(i)})$

---

pled from the BC policy) that maximizes the learned value function:

$$\pi(s_t) \overset{d}{:=} \underset{\{a_{t:t+n-1}^{(i)}\}_{i=1}^{N} \sim \pi^\beta(a_{t:t+n-1}|s_t)}{\arg\max} Q(s_t, a_{t:t+n-1}^{(i)}), \tag{4}$$

where $\pi^\beta(a_{t:t+n-1} \mid s_t) : \mathcal{S} \to \Delta(\mathcal{A}^n)$ denotes an action-chunk flow BC policy, $Q(s_t, a_{t:t+n-1}) : \mathcal{S} \times \mathcal{A}^n \to \mathbb{R}$ denotes an action-chunk value function, and $\overset{d}{=}$ denotes equality in distribution.

Compared to Gaussian policies, the flow-based behavior policy better models multi-modal action distributions, allowing us to sample action chunks that stay in-distribution, which obviates the need for an additional uncertainty penalization mechanism. Moreover, rejection sampling is generally more robust to hyperparameters (Zhou et al., 2025; Park et al., 2025b), making our method simpler and easier to tune than other alternatives, which may require tuning an uncertainty penalization coefficient for each task.

## 4.2 PRACTICAL ALGORITHM

Based on the idea discussed in the previous section, we now describe the full details of our method for scalable offline model-based RL, which we call **Model-Based RL with Action Chunks (MAC)**. MAC consists of the following components: an action-chunk dynamics model $p_\psi$, an action-chunk reward model $r_\psi$, a flow action-chunk policy $\pi_\theta$, and value functions $V_\varphi$ and $Q_\varphi$. For notational simplicity, we override the symbols $\psi$, and $\varphi$ to denote all model-, and value-related parameters, respectively. Moreover, we denote $\boldsymbol{a}_t \in \mathcal{A}^n$ to be the action chunk $a_{t:t+n}$, and $\boldsymbol{r}_t$ to be the sum of discounted rewards for $n$ steps $\sum_{i=0}^{n-1} \gamma^i r_{t+i}$.

**Action-chunk dynamics and reward models.** For dynamics modeling, we minimize the following losses to train a deterministic action-chunk dynamics model $p_\psi(s_t, a_t) : \mathcal{S} \times \mathcal{A}^n \to \mathcal{S}$ and an

action-chunk reward model $r_\psi(s_t, \boldsymbol{a}_t) : \mathcal{S} \times \mathcal{A}^n \to \mathbb{R}$:

$$L^{\text{dyn}}(\psi) = \mathbb{E}_{(s_t, a_t, \cdots, s_{t+n}) \sim \mathcal{D}} \left[ \| p_\psi(s_t, \boldsymbol{a}_t) - s_{t+n} \|_2^2 \right], \tag{5}$$

$$L^{\text{rew}}(\psi) = \mathbb{E}_{(s_t, a_t, \cdots, s_{t+n}) \sim \mathcal{D}} \left[ \| r_\psi(s_t, \boldsymbol{a}_t) - \boldsymbol{r}_t \|_2^2 \right], \tag{6}$$

where trajectory chunks are uniformly sampled from the offline dataset. The dynamics function $p_\psi$ is modeled by a deterministic multi-layer perceptron (MLP). While we found this to be sufficient in our benchmark environments, we note that it is possible to replace the MLP with an expressive flow model (as in our policy) in stochastic or partially observable environments.

**Flow action-chunk policies.** For the actor, we employ rejection sampling using a behavioral flow action-chunk policy, as described in Section 4.1. To train a flow BC policy, we train a state-dependent velocity field $v_\theta : \mathbb{R} \times \mathcal{S} \times \mathcal{A}^n \to \mathcal{A}^n$, with the flow-matching loss (Equation (2)):

$$L^{\text{flow}}(\theta) = \mathbb{E}_{\substack{z \sim \mathcal{N}(0, I_{nd}),\ (s_t, \boldsymbol{a}_t) \sim \mathcal{D}, \\ u \sim \text{Unif}([0,1]),\ a_z = (1-u)z + uz}} \left[ \| v_\theta(u, s_t, a_z) - (\boldsymbol{a}_t - z) \|_2^2 \right]. \tag{7}$$

We define $\pi_\theta(s_t, z) \in \mathcal{A}^n$ as the destination of the induced flow at $u = 1$ when starting with $(s_t, z)$ at $u = 0$ and following the velocity field $v_\theta$. Then, by sampling multiple noises $z \sim \mathcal{N}(0, I_{nd})$ and computing $\pi_\theta(s_t, z)$, we can obtain behavioral action-chunk samples, which are then used for rejection sampling (Equation (4)) along with a learned value function (described in the "Value learning" section below).

One issue with this rejection sampling framework is speed. To compute a single action chunk using Equation (4), we need $NF$ queries of the velocity field $v_\theta$, where $N$ is the number of samples and $F$ is the number of flow steps in the Euler method [1]. For example, with $N = 8$ and $F = 10$, we need to query the velocity field 80 times to sample a single action chunk. This is particularly prohibitive in model-based RL, as we need to sample multiple imaginary rollouts during training in batches, unlike methods that employ rejection sampling only at test time (Hansen-Estruch et al., 2023; Park et al., 2025b; Zhou et al., 2025).

To address this issue, we train an additional *one-step*[2] flow policy that directly predicts the output of the ODE flow policy. Specifically, we train a one-step MLP action-chunk policy $\pi_\omega(s_t, z)$ : $\mathcal{S} \times \mathcal{A}^n \to \mathcal{A}^n$ parameterized by $\omega$, with the following flow distillation loss (Park et al., 2025c):

$$L^{\text{distill}}(\omega) = \mathbb{E}_{s_t \sim \mathcal{D},\ z \sim \mathcal{N}(0, I_{dn})} \left[ \| \pi_\omega(s_t, z) - [\pi_\theta(s_t, z)]_\times \|_2^2 \right], \tag{8}$$

where $[\cdot]_\times$ denotes the "stop gradient" operation.

Unlike the ODE policy, $\pi_\omega$ only requires a single network call to produce an action chunk, reducing the number of queries from $NF$ to $N$ for rejection sampling in Equation (4). This substantially reduces both the training and inference cost of MAC.

**Value learning.** In MAC, value functions are trained from on-policy model rollouts (i.e., imaginary trajectories). To train value functions, we first generate $M$ imaginary (action-chunk) trajectories of length $H$,

$$\mathcal{D}^{\text{img}} = \{(s_t^{(i)}, \hat{\boldsymbol{a}}_t^{(i)}, \hat{\boldsymbol{r}}_t^{(i)}, \hat{s}_{t+n}^{(i)}, \hat{\boldsymbol{a}}_{t+n}^{(i)}, \hat{\boldsymbol{r}}_{t+n}^{(i)}, \dots, \hat{s}_{t+Hn}^{(i)})\}_{i=1}^M, \tag{9}$$

where $\hat{\boldsymbol{a}}_t$ denotes the action chunk $\hat{a}_{t:t+n}$ generated from the rejection sampling policy, and $\hat{\boldsymbol{r}}_t$ denotes the discounted sum of rewards $\sum_{i=0}^{n-1} \gamma^i r_{t+i}$ predicted from the reward model $r_\psi(\cdot | s_t, \boldsymbol{a}_t)$. Here, initial states $s_t^{(i)}$ are uniformly sampled from the dataset, and subsequent actions, rewards, and next states are synthesized by our rejection-sampling policy, reward model, and dynamics model, respectively, hence the hat notation.

After collecting $\mathcal{D}^{\text{img}}$, we update the value function $V_\varphi(s_t) : \mathcal{S} \to \mathbb{R}$ with the following loss:

$$L^V(\varphi) = \mathbb{E} \left[ \left( V_\varphi(\hat{s}_{t+kn}) - \sum_{i=k}^{H-1} \gamma^{(i-k)n} \hat{\boldsymbol{r}}_{t+in} - \gamma^{(H-k)n} V_{\bar{\varphi}}(\hat{s}_{t+Hn}) \right)^2 \right], \tag{10}$$

---

[1] We note that wall-clock training time heavily depends on $F$ than $N$, since rejection sampling can be parallelized, while flow sampling is not.

[2] We emphasize that "one-step" is different from "environment steps" in RL. Although the phrasing can be ambiguous, "one-step" (or "single-step") is the standard term for single-step distillation procedures (e.g., as used in Park et al. (2025c); Frans et al. (2025)). Because this terminology is already conventional, we would like to retain "one-step" for consistency with prior works.

where $\bar{\varphi}$ denotes exponentially averaged target parameters (Mnih et al., 2013), and the expectations are over $(s_t = \hat{s}_t, \hat{a}_t, \hat{r}_t, \ldots, \hat{s}_{t+Hn})$ uniformly sampled from $\mathcal{D}^{\text{img}}$ and $k$ uniformly sampled from $\{0, 1, \ldots, H-1\}$.

Finally, we train the action-chunk Q function $Q_\varphi(s_t, a_t) : \mathcal{S} \times \mathcal{A}^n \to \mathbb{R}$ for the rejection sampling with the following loss:

$$L^Q(\varphi) = \mathbb{E}_{s_t \sim \mathcal{D}} \left[ (Q_\varphi(s_t, \hat{a}_t) - \hat{r}_t - \gamma^n [V_\varphi(\hat{s}_{t+n})]_\times)^2 \right]. \tag{11}$$

We do not reuse $\mathcal{D}^{\text{img}}$ after performing one gradient update of value functions; *i.e.*, we generate new model rollouts every epoch. We provide a pseudocode for MAC in Algorithm 1.

**Notes on hyperparameters.** While MAC has several learnable components, MAC is comparatively **easier to tune** the hyperparameters than prior methods in our experiments. In particular, we use the same horizon hyperparameters of $(n, H) = (10, 10)$ for **all** tasks considered in this work. We also use the same number ($N = 32$) of samples for flow rejection sampling during evaluation across all tasks. That is, we can use the hyperparameters across all tasks, while prior model-based baselines (e.g., MOBILE, MOPO) requires task-specific rollout horizons and uncertainty penalties to remain stable. See Appendix A for the full details.

## 5 EXPERIMENTS

Now, we empirically evaluate the performance of MAC through a series of experiments. Our main research question is how well MAC scales to *long-horizon* tasks compared to previous offline model-based RL approaches, which we answer in Section 5.1. Then, we compare MAC with previous methods on standard offline RL benchmark tasks to assess its effectiveness as a general offline RL algorithm (Section 5.2). Finally, we provide several analyses and ablation studies to understand the importance of each component of MAC (Section 5.3). In our experiments, we use four random seeds (unless otherwise mentioned) and report standard deviations in tables and 95% confidence intervals in plots. In tables, we highlight numbers that are above or equal to 95% of the best performance.

### 5.1 EXPERIMENTS ON LARGE-SCALE, LONG-HORIZON TASKS

We first study the horizon scalability of MAC by evaluating it on large-scale, long-horizon benchmark tasks.

**Tasks and datasets.** To assess the scalability limits of each algorithm, we employ three highly challenging, long-horizon simulated robotic tasks used in the work by Park et al. (2025b) modified from OGBench (Park et al., 2025a): `humanoidmaze-giant`, `cube-octuple`, and `puzzle-4x5`. These tasks are not just long-horizon but also goal-conditioned (*i.e.*, the agent must reach any goal states given at test time), requiring complex, *multi-task* reasoning over a long episode. They present a variety of control challenges from high-dimensional humanoid navigation to complex object manipulation and combinatorial puzzle solving. The hardest task in each environment requires 700–3000 environment steps and 8–20 different atomic motions to complete. In addition to these long-horizon tasks, we also evaluate methods on shorter-horizon variants in each category (*i.e.*, `humanoidmaze-medium`, `cube-double`, and `puzzle-3x3`) to examine each method's ability to handle different horizon lengths.

For datasets, we mainly employ the 100M-transition datasets provided by Park et al. (2025b). These large-scale datasets are collected in a task-agnostic manner (*e.g.*, trajectories consisting of random atomic motions), meaning that the agent must understand the dynamics and stitch different parts of trajectories to achieve test-time tasks.

**Methods.** We mainly compare MAC against six previous model-based RL methods across diverse categories, including flat and hierarchical, and actor-critic and planning approaches.

Among standard model-based RL approaches, we consider MOPO, MOBILE, LEQ, and F-MPC. MOPO (Yu et al., 2020) and MOBILE (Sun et al., 2023) are Dyna-style methods (*i.e.*, ones that generate imaginary rollouts, augment the dataset, and run off-policy RL) based on different uncertainty penalization techniques. LEQ (Park & Lee, 2025) is a model-based actor-critic method based on conservative return estimation. F-MPC is a flow-based variant of D-MPC (Zhou et al., 2025), which

Table 1: **Results on large-scale, long-horizon tasks.** MAC achieves the best performance among model-based RL algorithms.

| Environment | Model-Free | | | Seq. Modeling | | Model-Based | | | | |
|---|---|---|---|---|---|---|---|---|---|---|
| | GCIQL | n-SAC+BC | SHARSA | Diffuser | HD-DA | MOPO | MOBILE | LEQ | FMPC | MAC |
| humanoidmaze-medium-navigate-oraclerep-v0 | 55 ±1 | 98 ±2 | 95 ±2 | 0 ±0 | 0 ±0 | 27 ±5 | 23 ±3 | 0 ±0 | 18 ±5 | 36 ±2 |
| humanoidmaze-giant-navigate-oraclerep-v0 | 4 ±2 | 82 ±5 | 43 ±6 | 0 ±0 | 0 ±0 | 0 ±0 | 0 ±0 | 0 ±0 | 0 ±0 | 0 ±0 |
| cube-double-play-oraclerep-v0 | 74 ±3 | 32 ±20 | 95 ±3 | 1 ±1 | 2 ±1 | 25 ±12 | 15 ±3 | 0 ±0 | 37 ±13 | 100 ±1 |
| cube-octuple-play-oraclerep-v0 | 0 ±0 | 0 ±0 | 19 ±3 | 0 ±0 | 0 ±0 | 0 ±0 | 0 ±0 | 0 ±0 | 0 ±0 | 30 ±6 |
| puzzle-3x3-play-oraclerep-v0 | 98 ±3 | 91 ±2 | 100 ±0 | 1 ±1 | 1 ±1 | 19 ±2 | 15 ±5 | 1 ±1 | 12 ±6 | 100 ±0 |
| puzzle-4x5-play-oraclerep-v0 | 20 ±1 | 19 ±4 | 91 ±4 | 0 ±0 | 0 ±0 | 0 ±0 | 0 ±0 | 1 ±3 | 0 ±0 | 99 ±3 |

trains an action-chunk dynamics model as in our method, but performs planning (based on a behavioral Monte Carlo value function) instead of training an on-policy value function with actor-critic.

Among sequence modeling approaches, we consider Diffuser and HD-DA. Diffuser (Janner et al., 2022) models trajectories with diffusion (Ho et al., 2020) for planning, and HD-DA (Chen et al., 2024) extends Diffuser using hierarchical models and high-level planning to handle long horizons.

For reference, we additionally consider three performant model-free RL algorithms as well: IQL, $n$-SAC+BC, and SHARSA. IQL (Kostrikov et al., 2022) is a standard model-free offline RL algorithm based on in-sample value learning. $n$-SAC+BC (Park et al., 2025b) is a behavior-regularized offline RL method that employs $n$-step returns to handle long horizons. SHARSA (Park et al., 2025b) is a state-of-the-art offline RL algorithm designed for long-horizon tasks that employs hierarchical policies and flow rejection sampling.

### 5.1.1 RESULTS

We present the main comparison results on six tasks in Table 1. The results suggest that MAC achieves the best performance across all settings among model-based RL algorithms. In particular, none of the previous model-based RL approaches achieves non-trivial performance on three long-horizon tasks. This is likely because they either use single-step models, which suffer from error accumulation (see Figure 2), or are based on planning, which is insufficient to perform full-fledged long-horizon dynamic programming. Moreover, even compared to state-of-the-art model-free RL approaches (*e.g.*, SHARSA), MAC achieves the best performance on four out of six tasks, especially on long-horizon manipulation tasks (`cube-octuple` and `puzzle-4x5`).

**Negative results.** Despite its strength on manipulation tasks, MAC, as well as all other model-based RL approaches, struggles on long-horizon robotic locomotion tasks (*e.g.*, `humanoidmaze-giant`). This is a widely known phenomenon; prior works (Chitnis et al., 2024; Park & Lee, 2025) have also found that model-based RL particularly struggles in similar robotic maze navigation environments (*e.g.*, `antmaze-large` in D4RL (Fu et al., 2020)). We believe this is mainly due to the difficulties in modeling *contact-rich* dynamics in locomotion domains, where dynamics tend to be highly erratic due to discontinuities, resulting in severe model error accumulation. While MAC's action-chunk dynamics model does mitigate this issue to some extent, leading to the best performance among model-based RL approaches (Table 1), it is not sufficient to fully close the gap between model-free and model-based approaches on these locomotion tasks. We believe this issue may be addressed by more expressive generative models or latent dynamics models, which we leave for future work.

### 5.2 EXPERIMENTS ON STANDARD BENCHMARKS

Next, we evaluate MAC on standard, reward-based benchmark tasks to assess its ability to serve as a general offline RL algorithm under limited data.

**Tasks and datasets.** We employ 25 single-task manipulation tasks from five environments in OG-Bench (Park et al., 2025a): `cube-{single, double}`, `scene`, and `puzzle-{3x3, 4x4}`. Unlike in Section 5.1, these tasks are reward-based (*i.e.*, not goal-conditioned), where the agent gets a reward according to the progress of the task. We use the 1M-sized `play` datasets given by the benchmark. We report the average success rate across 5 tasks for each environment.

**Methods.** For model-based approaches, we consider the four standard model-based RL algorithms used in Section 5.1. Additionally, we consider four standard, performant model-free RL algorithms used in the work by Park et al. (2025c): IQL (Kostrikov et al., 2022), ReBRAC (Tarasov et al.,

Table 2: **Results on standard reward-based benchmark tasks.** MAC achieves the best performance among both model-based and model-free RL algorithms.

| | Model-Free | | | | Model-Based | | | | |
|---|---|---|---|---|---|---|---|---|---|
| Environment | IQL | ReBRAC | IDQL | FQL | MOPO | MOBILE | LEQ | FMPC | MAC |
| cube-single-play-v0 (5 tasks) | $83_{\pm9}$ | $91_{\pm5}$ | $95_{\pm4}$ | $96_{\pm3}$ | $12_{\pm4}$ | $81_{\pm8}$ | $0_{\pm0}$ | $9_{\pm5}$ | $99_{\pm2}$ |
| cube-double-play-v0 (5 tasks) | $7_{\pm11}$ | $12_{\pm17}$ | $15_{\pm17}$ | $29_{\pm21}$ | $1_{\pm1}$ | $1_{\pm2}$ | $0_{\pm0}$ | $3_{\pm2}$ | $53_{\pm4}$ |
| scene-play-v0 (5 tasks) | $28_{\pm36}$ | $41_{\pm37}$ | $46_{\pm44}$ | $56_{\pm45}$ | $6_{\pm8}$ | $8_{\pm4}$ | $0_{\pm0}$ | $4_{\pm4}$ | $97_{\pm4}$ |
| puzzle-3x3-play-v0 (5 tasks) | $9_{\pm13}$ | $21_{\pm38}$ | $10_{\pm21}$ | $30_{\pm31}$ | $20_{\pm0}$ | $12_{\pm9}$ | $10_{\pm7}$ | $1_{\pm1}$ | $20_{\pm0}$ |
| puzzle-4x4-play-v0 (5 tasks) | $7_{\pm4}$ | $14_{\pm8}$ | $29_{\pm13}$ | $17_{\pm10}$ | $0_{\pm0}$ | $0_{\pm0}$ | $0_{\pm0}$ | $0_{\pm0}$ | $78_{\pm13}$ |

2023), IDQL (Hansen-Estruch et al., 2023), and FQL (Park et al., 2025c). Among them, FQL is a state-of-the-art model-free offline RL method on these tasks.

### 5.2.1 RESULTS

Table 2 summarizes the comparison results on 25 standard benchmark tasks. The results show that MAC achieves the best average performance on four out of five environments. Notably, MAC achieves substantially better performance than all other methods especially on (relatively) long-horizon environments, such as cube-double, scene, and puzzle-4x4. MAC also outperforms state-of-the-art model-free RL algorithms, showing the promise of offline model-based RL in manipulation domains.

### 5.3 Q&AS

In this section, we discuss and analyze the components of MAC through the following Q&As.

**Q: Do action chunks actually mitigate error accumulation?**

**A:** Our main motivation for using action chunking is to reduce error accumulation in autoregressive trajectory generation. However, one might question whether it is actually the case, given that increasing the action chunk length also increases the difficulty of learning the model. To examine this, we analyze how the chunk length affects model errors. Specifically, we train dynamics models with action chunk lengths of $\{1, 5, 10, 25\}$ and measure their mean squared prediction errors along a length-100 dataset trajectory in puzzle-4x5. Figure 2

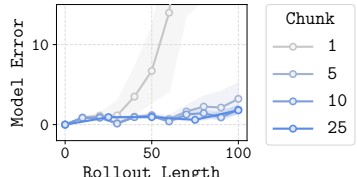

Figure 2: **Action chunking reduces model errors.**

presents the result, suggesting that longer action chunks indeed substantially mitigate error accumulation. Notably, the errors from a standard one-step model diverge over time, substantiating the necessity of multi-step prediction for long-horizon tasks.

**Q: How does the action chunk length affect performance?**

**A:** To answer this question, we train MAC with four action chunk lengths ($\{1, 5, 10, 25\}$) on one short-horizon and one long-horizon task (cube-double and cube-octuple, respectively) used in Section 5.1. As shown in Figure 3, action chunking with an appropriate chunk size can substantially improve performance on both tasks. Notably, while cube-double can still be partially solved without action chunking,

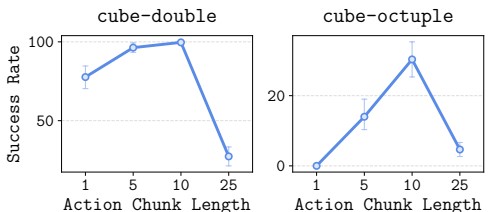

Figure 3: **Action chunk length vs. performance.**

cube-octuple cannot be solved at all without it. This demonstrates that action chunking is crucial especially on long-horizon tasks. However, Figure 3 also shows that too long action chunks can degrade performance, mainly due to the difficulty of open-loop multi-step future prediction.

**Q: How important is flow rejection sampling?**

**A:** Another key feature of MAC is its use of flow rejection sampling. To understand the importance of this component, we conduct an ablation study of MAC by using (1) a Gaussian ("Gau") action-chunk policy instead of a flow policy, and (2) gradient-based policy extraction (one-step distil-

Table 3: **Ablation study of MAC.**

| Task | MAC (Gau) | MAC (FQL) | MAC |
|---|---|---|---|
| cube-single-play-v0 | $2_{\pm3}$ | $77_{\pm21}$ | $100_{\pm0}$ |
| cube-double-play-v0 | $0_{\pm0}$ | $2_{\pm3}$ | $50_{\pm12}$ |
| scene-play-v0 | $0_{\pm0}$ | $40_{\pm47}$ | $100_{\pm0}$ |
| puzzle-3x3-play-v0 | $0_{\pm0}$ | $0_{\pm0}$ | $0_{\pm0}$ |
| puzzle-4x4-play-v0 | $0_{\pm0}$ | $23_{\pm13}$ | $85_{\pm14}$ |

lation from FQL (Park et al., 2025c)) instead of
rejection sampling. We present the ablation results on the default tasks for five reward-based environments used in Table 3. The results indicate that the use of expressive flow matching is crucial for MAC, and that rejection sampling generally yields better performance on most tasks.

## 6 CLOSING REMARKS

In this work, we introduced MAC as a model-based actor-critic algorithm that combines an action-chunk policy and an action-chunk model. MAC enables generating imaginary autoregressive rollouts up to 100 steps, achieving the best performance among model-based RL approaches on challenging, long-horizon tasks.

We now revisit the initial promise of this paper. In Section 1, we motivated offline model-based RL as a promising alternative to offline model-free RL in terms of horizon scalability. Our answer is (at least partially) affirmative: on a variety of long-horizon manipulation tasks, we show that MAC does outperform state-of-the-art model-free RL algorithms (Table 1). However, as discussed in Section 5.1, even the best model-based RL algorithm (MAC) underperforms on contact-rich locomotion tasks (*e.g.*, humanoidmaze), suggesting room for improvement in sequential dynamics modeling. Moreover, value learning can become challenging when chunk sizes are very large (Figure 3), and rejection sampling may limit performance on low-quality datasets (e.g., random datasets) as the behavioral policy cannot provide useful guidance for policy extraction. We believe that incorporating more advanced modeling techniques and policy extraction techniques could address these limitations, which we leave for future work.

## REPRODUCIBILITY STATEMENT

For the reproducibility of our work, we provide the code of MAC in https://github.com/kwanyoungpark/MAC. We fully describe the experimental details and hyperparameters to reproduce the results for our method and baselines in Appendix A.

## ACKNOWLEDGEMENTS

Kwanyoung Park and Seohong Park are partly supported by the Korea Foundation for Advanced Studies (KFAS). This research used the Savio computational cluster resource provided by the Berkeley Research Computing program at UC Berkeley. This research was partly supported by ONR N00014-22-1-2773 and the National Research Foundation of Korea (NRF) grants funded by the Korean Government (MSIT) (RS-2024-00333634, RS-2025-25396144, RS-2025-25448259).

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

# A    EXPERIMENTAL DETAILS

We implement MAC on top of the codebase of Park et al. (2025b). Each experiment takes approximately 2 days for large-scale benchmarks, and around 3 hours for single-task benchmarks on a single A5000 GPU. Please refer to Appendix A.2 for detailed time measures.

## A.1    IMPLEMENTATION DETAILS

**Network architectures.** We follow the setup of the work by Park et al. (2025c;b), using 4-layer MLPs with layer normalization (Ba et al., 2016) for all neural networks (the policy, critic, dynamics model, and reward model). For large-scale benchmarks, we parameterize the reward model and the terminal model using a single success prediction network $f_\psi(s_t, a_t)$, where termination is calculated as $\mathbb{1}(f_\psi(s_t, a_t) > 0.5)$ and reward is calculated as $\mathbb{1}(f_\psi(s_t, a_t) > 0.5) - 1$. For reward-based tasks, we use a reward model $r_\psi(s_t, a_t)$ without termination.

**Accelerating rejection sampling during training.** To improve training time, we use different numbers of samples for rejection sampling during training and evaluation (which we call $N_{\text{train}}$ and $N_{\text{test}}$. Specifically, we use $N_{\text{train}} = 8$ during training (except in `puzzle-4x5`, where a larger $N_{\text{train}} = 32$ was necessary due to the BC policy branching over 20 possible actions) and $N_{\text{test}} = 32$ at test time.

**Implementation details for the compared methods.** We implement MOPO, MOBILE, and LEQ in our codebase. For MOPO, epistemic uncertainty is estimated as the maximum standard deviation across ensemble members (Yu et al., 2020). For LEQ, we omit dataset expansion, which we found to have a negligible impact in our benchmarks. We use 5 dynamics model ensembles for all methods and disable early stopping and validation filtering when training the model, as we found they are unreliable on large-scale datasets (training and validation metrics are nearly identical in these settings).

For D-MPC (Zhou et al., 2025), we implement the flow variant of D-MPC (F-MPC) in our codebase. Specifically, we train a flow BC policy $\pi(a_{t:t+n-1} \mid s_t)$ and a flow dynamics model $p_\psi(s_{t+1:t+n} \mid s_t, a_{t:t+n-1})$ instead of using diffusion models. For reward-based benchmarks, we calculate the return-to-go as $G_t = \sum_{t'=t}^{T} r_{t'}$ without discounts, as in the original paper. For goal-conditioned (large-scale) benchmarks, we similarly define the return-to-go without discounts for the goal-conditioned tasks as $G_t = \mathbb{1}(g \in \{s_t, \cdots, s_T\})$. Unlike the original architecture, we do not use history conditioning and transformers (as all tasks are Markovian) and use the same MLP architecture as other methods for a fair comparison.

For sequence modeling approaches (Diffuser and HD-DA), we follow the official implementation for D4RL's `maze2d` environment (Fu et al., 2020), and adjust the maximum length of the trajectory generation and the number of diffusion steps (of the high-level policy for HD-DA) to be the maximum length of the environment (*e.g.*, $H = 4000$ for `humanoidmaze-giant`). We re-plan the trajectory every 100 steps, as we found that this is necessary to achieve a non-zero performance on long-horizon tasks, unlike in the `maze2d` benchmark.

For other model-free methods, we use the implementations by Park et al. (2025b) and Park et al. (2025a). We also take the results from these papers for the corresponding methods.

**Implementation details for ablation experiments.** For the ablation study on the action-chunk length, we fix the horizon length $H$ to 10 and only change the action-chunk length $n \in \{1, 5, 10, 25\}$. For MAC (Gau) of the ablation study on flow rejection sampling, we parameterize the action-chunk policy with $a_t = \tanh(x_t)$, where $x_t \sim \mathcal{N}(\mu_\theta(s_t), \sigma_\theta^2(s_t))$.

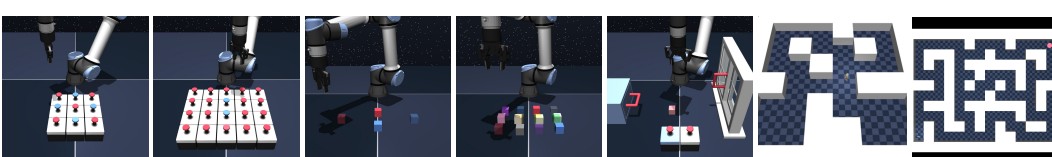

Figure 4: **OGBench tasks.**

## A.2 TRAINING TIME

We report the average training time and inference time for single-task and multi-task experiments in A5000 for MAC and prior MBRL methods in the table below. MAC trains in around 3 hours for a single task and 55 hours for multi-task experiments, which is 1.2 - 2.2 times longer than other methods. Inference speed of MAC is similar or 1.5 times longer than other methods. All models use identical architecture sizes across methods.

Table 4: **Training time of MAC and prior MBRL methods.**

| Training time (hours) | MOPO | MOBILE | FMPC | LEQ (H=5) | MAC |
|---|---|---|---|---|---|
| **Single-task** | 1.4 | 2.6 | 1.7 | 1.6 | 3.1 |
| **Multi-task** | 25.1 | 36.7 | 28.4 | 25.2 | 55.5 |

Table 5: **Inference time of MAC and prior MBRL methods.**

| Inference time (ms) | MOPO | MOBILE | FMPC | LEQ | MAC |
|---|---|---|---|---|---|
| **Single-task** | 1.8 | 1.8 | 2.3 | 1.6 | 2.5 |
| **Multi-task** | 5.1 | 5.2 | 7.3 | 5.2 | 7.2 |

## A.3 HYPERPARAMETERS

**Shared hyperparameters.** Here, we report shared hyperparameters for MAC, MOPO, MOBILE, and all model-free baselines. The hyperparameters for goal-conditioned tasks are presented in Table 6, and those for reward-based tasks are in Table 7. We note that these hyperparameter configurations mostly follow those of SHARSA (Park et al., 2025b) for multi-task experiments, and FQL (Park et al., 2025c) for single-task experiments.

Table 6: **Shared hyperparameters for large-scale benchmark tasks.**

| Hyperparameters | Value |
|---|---|
| Gradient steps | 2M |
| Learning rate | $3 \times 10^{-4}$ |
| Optimizer | Adam (Kingma & Ba, 2015) |
| Batch size | 1024 |
| MLP size | $[1024, 1024, 1024, 1024]$ |
| Actor $(p_{\text{cur}}^{\mathcal{D}}, p_{\text{geom}}^{\mathcal{D}}, p_{\text{traj}}^{\mathcal{D}}, p_{\text{rand}}^{\mathcal{D}})$ ratio | $(0, 1, 0, 0)$ (cube) |
| | $(0, 0.5, 0, 0.5)$ (puzzle) |
| | $(0, 0, 1, 0)$ (humanoidmaze) |
| Value $(p_{\text{cur}}^{\mathcal{D}}, p_{\text{geom}}^{\mathcal{D}}, p_{\text{traj}}^{\mathcal{D}}, p_{\text{rand}}^{\mathcal{D}})$ ratio | $(0.2, 0, 0.5, 0.3)$ |

Table 7: **Shared hyperparameters for reward-based benchmark tasks.**

| Hyperparameters | Value |
|---|---|
| Gradient steps | 1M |
| Learning rate | $3 \times 10^{-4}$ |
| Optimizer | Adam (Kingma & Ba, 2015) |
| Batch size | 256 |
| MLP size | $[512, 512, 512, 512]$ |

**MAC hyperparameters.** We report the hyperparameters for our method in Table 8. Note that MAC uses the same $(n, H, N_{\text{train}}, N_{\text{test}}) = (10, 10, 8, 32)$ across all tasks, except for puzzle-4x5, where using $N_{\text{train}} = 32$ during training is important as the BC policy has 20 possible branches.

Accordingly, only in `puzzle-4x5`, we decrease the hidden dimensionality of the networks to 256 to compensate for the increased training time from $N_{\text{train}} = 32$.

Table 8: **Hyperparameters of MAC.**

| Hyperparameters | Value |
|---|---|
| Learning rate | $3 \times 10^{-4}$ |
| Optimizer | Adam |
| Nonlinearity | GELU (Hendrycks & Gimpel, 2016) |
| Layer normalization | True |
| Target network update rate | 0.005 |
| Discount factor $\gamma$ | 0.999 |
| Flow steps | 10 |
| $N_{\text{train}}$ | 8 (default), 32 (`puzzle-4x5`) |
| $N_{\text{test}}$ | 32 |
| Rollout length $H$ | 10 |
| Action-chunk size $n$ | 10 |

**Hyperparameters for baselines.** We report the optimal hyperparameters of all baselines for goal-conditioned experiments in Table 9 and reward-based experiments in Table 11.

For MOPO and MOBILE, we perform a hyperparameter sweep over rollout lengths $H \in \{1, 5, 10\}$ and penalty coefficients $\beta \in \{0.1, 0.5, 1.0, 2.0, 3.0, 5.0\}$, where $H$ denotes the model rollout horizon and $\beta$ is the penalization coefficient for model uncertainty or Bellman inconsistency, respectively. We note that reducing the MBPO loop's model batch ratio $f$ from 0.95 to $f \in \{0.5, 0.25\}$ is crucial for training on long-horizon tasks, as also noted by Park & Lee (2025).

For LEQ, we search over rollout lengths $H \in \{1, 5, 10\}$ and expectiles $\tau \in \{0.1, 0.3, 0.5\}$, where the expectile $\tau$ controls the degree of conservatism for critic and policy learning.

For model-free methods in large-scale benchmarks, we follow the list of hyperparameters to search over in the work by Park et al. (2025b). For SHARSA, we searched over $n \in \{25, 50\}$. For n-SAC+BC, we search over $n \in 10, 25, 50$ and regularization coefficients $\alpha \in \{0.01, 0.03, 0.1, 0.3\}$. For GCIQL, we follow (Park et al., 2024) and extract policies with DDPG+BC, searching over $\alpha \in \{0.003, 0.01, 0.03, 0.1, 0.3, 1.0, 3.0\}$.

We denote "N/A" in the tables if a method achieves zero performance across all hyperparameters tested in our sweep. If not specified, all other hyperparameters follow the defaults provided in the original papers.

Table 9: **Hyperparameters for baselines for large-scale benchmark tasks.**

| Environment | MOPO $(H, \beta, f)$ | MOBILE $(H, \beta, f)$ | LEQ $(H, \tau)$ |
|---|---|---|---|
| `cube-double-play-v0` | $(10, 1.0, 0.25)$ | $(5, 0.5, 0.5)$ | N/A |
| `cube-octuple-play-v0` | N/A | N/A | N/A |
| `humanoidmaze-medium-navigate-v0` | $(1, 0.5, 0.5)$ | $(1, 1.0, 0.5)$ | N/A |
| `humanoidmaze-giant-navigate-v0` | N/A | N/A | N/A |
| `puzzle-3x3-play-v0` | $(5, 5.0, 0.5)$ | $(10, 3.0, 0.25)$ | $(1, 0.1)$ |
| `puzzle-4x5-play-v0` | N/A | N/A | $(1, 0.1)$ |

Table 10: **Hyperparameters for baselines for reward-based benchmark tasks.**

| Environment | MOPO $(H, \beta, f)$ | MOBILE $(H, \beta, f)$ | LEQ $(H, \tau)$ |
|---|---|---|---|
| `cube-single-play-v0` | $(10, 2.0, 0.25)$ | $(10, 5.0, 0.25)$ | N/A |
| `cube-double-play-v0` | $(10, 1.0, 0.25)$ | N/A | N/A |
| `scene-play-v0` | $(10, 2.0, 0.25)$ | N/A | N/A |
| `puzzle-3x3-play-v0` | N/A | N/A | N/A |
| `puzzle-4x4-play-v0` | N/A | N/A | N/A |

**Hyperparameters for ablation studies.** We report the optimal hyperparameters for the ablated variants of MAC: one that replaces the flow policy with a Gaussian policy ("Gau"), and another that replaces rejection sampling with FQL's one-step distillation ("FQL"). For the Gaussian policy variant, we reuse the same hyperparameters as our main method. For the FQL variant, we search over the behavior cloning coefficients $\alpha \in \{0.1, 0.3, 1.0, 3.0\}$.

Table 11: **Hyperparameters for ablation experiments.**

| Environment | MAC (FQL) ($\alpha$) |
|---|---|
| cube-single-play-v0 | 1.0 |
| cube-double-play-v0 | 0.3 |
| scene-play-v0 | 1.0 |
| puzzle-3x3-play-v0 | 1.0 |
| puzzle-4x4-play-v0 | 1.0 |

# B    COMPLETE NUMERICAL RESULTS

For completeness, we provide the full per-task results for large-scale, long-horizon environments and reward-based environments in Table 12 and Table 13 (corresponding to Table 1 and Table 2). The results are averaged over 4 seeds and we report the standard deviations for each tasks. We highlight the numbers that are above or equal to 95% of the best performance.

Table 12: **Complete results for large-scale experiments.**

| Environment | Task | Model-Free | | | Seq. Modeling | | Model-Based | | | | |
|---|---|---|---|---|---|---|---|---|---|---|---|
| | | GCIQL | n-SAC+BC | SHARSA | Diffuser | HD-DA | MOPO | MOBILE | LEQ | FMPC | MAC |
| humanoidmaze-medium-navigate-oraclerep-v0 | task1 | $82_{\pm3}$ | $97_{\pm4}$ | $95_{\pm6}$ | $0_{\pm0}$ | $0_{\pm0}$ | $48_{\pm27}$ | $38_{\pm14}$ | $0_{\pm0}$ | $27_{\pm9}$ | $67_{\pm12}$ |
| | task2 | $95_{\pm6}$ | $100_{\pm0}$ | $100_{\pm0}$ | $0_{\pm0}$ | $0_{\pm0}$ | $85_{\pm21}$ | $75_{\pm15}$ | $0_{\pm0}$ | $22_{\pm11}$ | $87_{\pm9}$ |
| | task3 | $0_{\pm0}$ | $98_{\pm3}$ | $100_{\pm0}$ | $0_{\pm0}$ | $0_{\pm0}$ | $0_{\pm0}$ | $0_{\pm0}$ | $0_{\pm0}$ | $18_{\pm3}$ | $7_{\pm0}$ |
| | task4 | $0_{\pm0}$ | $97_{\pm4}$ | $82_{\pm3}$ | $0_{\pm0}$ | $0_{\pm0}$ | $0_{\pm0}$ | $0_{\pm0}$ | $0_{\pm0}$ | $5_{\pm6}$ | $0_{\pm0}$ |
| | task5 | $98_{\pm3}$ | $98_{\pm3}$ | $100_{\pm0}$ | $0_{\pm0}$ | $0_{\pm0}$ | $0_{\pm0}$ | $0_{\pm0}$ | $0_{\pm0}$ | $20_{\pm11}$ | $22_{\pm14}$ |
| | overall | $55_{\pm1}$ | $98_{\pm2}$ | $95_{\pm2}$ | $0_{\pm0}$ | $0_{\pm0}$ | $27_{\pm5}$ | $23_{\pm3}$ | $0_{\pm0}$ | $18_{\pm5}$ | $36_{\pm2}$ |
| humanoidmaze-giant-navigate-oraclerep-v0 | task1 | $0_{\pm0}$ | $58_{\pm18}$ | $22_{\pm18}$ | $0_{\pm0}$ | $0_{\pm0}$ | $0_{\pm0}$ | $0_{\pm0}$ | $0_{\pm0}$ | $0_{\pm0}$ | $0_{\pm0}$ |
| | task2 | $10_{\pm7}$ | $87_{\pm8}$ | $43_{\pm22}$ | $0_{\pm0}$ | $0_{\pm0}$ | $0_{\pm0}$ | $0_{\pm0}$ | $0_{\pm0}$ | $0_{\pm0}$ | $0_{\pm0}$ |
| | task3 | $5_{\pm3}$ | $85_{\pm11}$ | $23_{\pm19}$ | $0_{\pm0}$ | $0_{\pm0}$ | $0_{\pm0}$ | $0_{\pm0}$ | $0_{\pm0}$ | $0_{\pm0}$ | $0_{\pm0}$ |
| | task4 | $2_{\pm3}$ | $82_{\pm11}$ | $40_{\pm14}$ | $0_{\pm0}$ | $0_{\pm0}$ | $0_{\pm0}$ | $0_{\pm0}$ | $0_{\pm0}$ | $0_{\pm0}$ | $0_{\pm0}$ |
| | task5 | $3_{\pm4}$ | $98_{\pm3}$ | $87_{\pm18}$ | $0_{\pm0}$ | $0_{\pm0}$ | $0_{\pm0}$ | $0_{\pm0}$ | $0_{\pm0}$ | $0_{\pm0}$ | $0_{\pm0}$ |
| | overall | $4_{\pm2}$ | $82_{\pm5}$ | $43_{\pm6}$ | $0_{\pm0}$ | $0_{\pm0}$ | $0_{\pm0}$ | $0_{\pm0}$ | $0_{\pm0}$ | $0_{\pm0}$ | $0_{\pm0}$ |
| cube-double-play-oraclerep-v0 | task1 | $100_{\pm0}$ | $67_{\pm32}$ | $100_{\pm0}$ | $6_{\pm3}$ | $6_{\pm3}$ | $42_{\pm18}$ | $50_{\pm12}$ | $0_{\pm0}$ | $73_{\pm14}$ | $100_{\pm0}$ |
| | task2 | $100_{\pm0}$ | $13_{\pm11}$ | $100_{\pm0}$ | $0_{\pm0}$ | $0_{\pm1}$ | $17_{\pm14}$ | $15_{\pm11}$ | $0_{\pm0}$ | $37_{\pm28}$ | $100_{\pm0}$ |
| | task3 | $100_{\pm0}$ | $37_{\pm23}$ | $100_{\pm0}$ | $0_{\pm0}$ | $0_{\pm1}$ | $18_{\pm15}$ | $7_{\pm5}$ | $0_{\pm0}$ | $43_{\pm23}$ | $100_{\pm0}$ |
| | task4 | $33_{\pm14}$ | $15_{\pm18}$ | $73_{\pm14}$ | $0_{\pm0}$ | $0_{\pm1}$ | $20_{\pm14}$ | $0_{\pm0}$ | $0_{\pm0}$ | $3_{\pm4}$ | $98_{\pm3}$ |
| | task5 | $38_{\pm16}$ | $28_{\pm33}$ | $100_{\pm0}$ | $0_{\pm1}$ | $1_{\pm0}$ | $30_{\pm13}$ | $5_{\pm3}$ | $0_{\pm0}$ | $30_{\pm16}$ | $100_{\pm0}$ |
| | overall | $74_{\pm3}$ | $32_{\pm20}$ | $95_{\pm3}$ | $1_{\pm1}$ | $2_{\pm1}$ | $25_{\pm12}$ | $15_{\pm3}$ | $0_{\pm0}$ | $37_{\pm13}$ | $100_{\pm1}$ |
| cube-octuple-play-oraclerep-v0 | task1 | $0_{\pm0}$ | $0_{\pm0}$ | $88_{\pm6}$ | $0_{\pm0}$ | $0_{\pm0}$ | $0_{\pm0}$ | $0_{\pm0}$ | $0_{\pm0}$ | $0_{\pm0}$ | $83_{\pm4}$ |
| | task2 | $0_{\pm0}$ | $0_{\pm0}$ | $5_{\pm10}$ | $0_{\pm0}$ | $0_{\pm0}$ | $0_{\pm0}$ | $0_{\pm0}$ | $0_{\pm0}$ | $0_{\pm0}$ | $20_{\pm9}$ |
| | task3 | $0_{\pm0}$ | $0_{\pm0}$ | $3_{\pm7}$ | $0_{\pm0}$ | $0_{\pm0}$ | $0_{\pm0}$ | $0_{\pm0}$ | $0_{\pm0}$ | $0_{\pm0}$ | $40_{\pm21}$ |
| | task4 | $0_{\pm0}$ | $0_{\pm0}$ | $0_{\pm0}$ | $0_{\pm0}$ | $0_{\pm0}$ | $0_{\pm0}$ | $0_{\pm0}$ | $0_{\pm0}$ | $0_{\pm0}$ | $5_{\pm6}$ |
| | task5 | $0_{\pm0}$ | $0_{\pm0}$ | $0_{\pm0}$ | $0_{\pm0}$ | $0_{\pm0}$ | $0_{\pm0}$ | $0_{\pm0}$ | $0_{\pm0}$ | $0_{\pm0}$ | $3_{\pm4}$ |
| | overall | $0_{\pm0}$ | $0_{\pm0}$ | $19_{\pm3}$ | $0_{\pm0}$ | $0_{\pm0}$ | $0_{\pm0}$ | $0_{\pm0}$ | $0_{\pm0}$ | $0_{\pm0}$ | $30_{\pm6}$ |
| puzzle-3x3-play-oraclerep-v0 | task1 | $100_{\pm0}$ | $95_{\pm6}$ | $100_{\pm0}$ | $3_{\pm2}$ | $4_{\pm4}$ | $93_{\pm9}$ | $77_{\pm23}$ | $3_{\pm7}$ | $25_{\pm15}$ | $100_{\pm0}$ |
| | task2 | $100_{\pm0}$ | $80_{\pm11}$ | $100_{\pm0}$ | $0_{\pm0}$ | $0_{\pm1}$ | $0_{\pm0}$ | $0_{\pm0}$ | $0_{\pm0}$ | $18_{\pm18}$ | $100_{\pm0}$ |
| | task3 | $98_{\pm3}$ | $93_{\pm5}$ | $100_{\pm0}$ | $0_{\pm0}$ | $0_{\pm0}$ | $0_{\pm0}$ | $0_{\pm0}$ | $0_{\pm0}$ | $8_{\pm8}$ | $100_{\pm0}$ |
| | task4 | $100_{\pm0}$ | $92_{\pm8}$ | $100_{\pm0}$ | $0_{\pm0}$ | $0_{\pm0}$ | $0_{\pm0}$ | $0_{\pm0}$ | $0_{\pm0}$ | $2_{\pm3}$ | $100_{\pm0}$ |
| | task5 | $93_{\pm13}$ | $95_{\pm6}$ | $100_{\pm0}$ | $0_{\pm0}$ | $0_{\pm0}$ | $0_{\pm0}$ | $0_{\pm0}$ | $0_{\pm0}$ | $7_{\pm9}$ | $100_{\pm0}$ |
| | overall | $98_{\pm3}$ | $91_{\pm2}$ | $100_{\pm0}$ | $1_{\pm1}$ | $1_{\pm1}$ | $19_{\pm2}$ | $15_{\pm5}$ | $1_{\pm1}$ | $12_{\pm6}$ | $100_{\pm0}$ |
| puzzle-4x5-play-oraclerep-v0 | task1 | $98_{\pm3}$ | $73_{\pm20}$ | $100_{\pm0}$ | $0_{\pm0}$ | $0_{\pm0}$ | $0_{\pm0}$ | $0_{\pm0}$ | $7_{\pm13}$ | $0_{\pm0}$ | $100_{\pm0}$ |
| | task2 | $0_{\pm0}$ | $15_{\pm18}$ | $100_{\pm0}$ | $0_{\pm0}$ | $0_{\pm0}$ | $0_{\pm0}$ | $0_{\pm0}$ | $0_{\pm0}$ | $0_{\pm0}$ | $100_{\pm0}$ |
| | task3 | $0_{\pm0}$ | $0_{\pm0}$ | $97_{\pm4}$ | $0_{\pm0}$ | $0_{\pm0}$ | $0_{\pm0}$ | $0_{\pm0}$ | $0_{\pm0}$ | $0_{\pm0}$ | $100_{\pm0}$ |
| | task4 | $0_{\pm0}$ | $8_{\pm8}$ | $92_{\pm6}$ | $0_{\pm0}$ | $0_{\pm0}$ | $0_{\pm0}$ | $0_{\pm0}$ | $0_{\pm0}$ | $0_{\pm0}$ | $100_{\pm0}$ |
| | task5 | $0_{\pm0}$ | $0_{\pm0}$ | $68_{\pm13}$ | $0_{\pm0}$ | $0_{\pm0}$ | $0_{\pm0}$ | $0_{\pm0}$ | $0_{\pm0}$ | $0_{\pm0}$ | $93_{\pm13}$ |
| | overall | $20_{\pm1}$ | $19_{\pm4}$ | $91_{\pm4}$ | $0_{\pm0}$ | $0_{\pm0}$ | $0_{\pm0}$ | $0_{\pm0}$ | $1_{\pm3}$ | $0_{\pm0}$ | $99_{\pm3}$ |

Table 13: **Complete results for reward-based experiments.**

| Environment | Task | Model-Free | | | | Model-Based | | | | |
|---|---|---|---|---|---|---|---|---|---|---|
| | | IQL | ReBRAC | IDQL | FQL | MOPO | MOBILE | LEQ | FMPC | MAC |
| cube-single-play-singletask-v0 | task1 | $88_{\pm 3}$ | $89_{\pm 5}$ | $95_{\pm 2}$ | $97_{\pm 2}$ | $12_{\pm 16}$ | $85_{\pm 22}$ | $0_{\pm 0}$ | $10_{\pm 9}$ | $100_{\pm 0}$ |
| | task2 | $85_{\pm 8}$ | $92_{\pm 4}$ | $96_{\pm 2}$ | $97_{\pm 2}$ | $10_{\pm 16}$ | $80_{\pm 12}$ | $0_{\pm 0}$ | $8_{\pm 8}$ | $100_{\pm 0}$ |
| | task3 | $91_{\pm 5}$ | $93_{\pm 3}$ | $99_{\pm 1}$ | $98_{\pm 2}$ | $15_{\pm 14}$ | $83_{\pm 17}$ | $0_{\pm 0}$ | $10_{\pm 9}$ | $98_{\pm 3}$ |
| | task4 | $73_{\pm 6}$ | $92_{\pm 3}$ | $93_{\pm 4}$ | $94_{\pm 3}$ | $2_{\pm 3}$ | $72_{\pm 19}$ | $0_{\pm 0}$ | $13_{\pm 9}$ | $98_{\pm 3}$ |
| | task5 | $78_{\pm 9}$ | $87_{\pm 8}$ | $90_{\pm 6}$ | $93_{\pm 3}$ | $20_{\pm 26}$ | $87_{\pm 19}$ | $0_{\pm 0}$ | $3_{\pm 4}$ | $97_{\pm 7}$ |
| | overall | $83_{\pm 9}$ | $91_{\pm 5}$ | $95_{\pm 4}$ | $96_{\pm 3}$ | $12_{\pm 4}$ | $81_{\pm 8}$ | $0_{\pm 0}$ | $9_{\pm 5}$ | $99_{\pm 2}$ |
| cube-double-play-singletask-v0 | task1 | $27_{\pm 5}$ | $45_{\pm 6}$ | $39_{\pm 19}$ | $61_{\pm 9}$ | $2_{\pm 3}$ | $7_{\pm 8}$ | $0_{\pm 0}$ | $15_{\pm 10}$ | $82_{\pm 15}$ |
| | task2 | $1_{\pm 1}$ | $7_{\pm 3}$ | $16_{\pm 10}$ | $36_{\pm 6}$ | $0_{\pm 0}$ | $0_{\pm 0}$ | $0_{\pm 0}$ | $0_{\pm 0}$ | $50_{\pm 12}$ |
| | task3 | $0_{\pm 0}$ | $4_{\pm 1}$ | $17_{\pm 8}$ | $22_{\pm 5}$ | $2_{\pm 3}$ | $0_{\pm 0}$ | $0_{\pm 0}$ | $0_{\pm 0}$ | $55_{\pm 10}$ |
| | task4 | $0_{\pm 0}$ | $1_{\pm 1}$ | $0_{\pm 1}$ | $5_{\pm 2}$ | $0_{\pm 0}$ | $0_{\pm 0}$ | $0_{\pm 0}$ | $0_{\pm 0}$ | $28_{\pm 8}$ |
| | task5 | $4_{\pm 3}$ | $4_{\pm 2}$ | $1_{\pm 1}$ | $19_{\pm 10}$ | $2_{\pm 3}$ | $0_{\pm 0}$ | $0_{\pm 0}$ | $0_{\pm 0}$ | $50_{\pm 9}$ |
| | overall | $7_{\pm 11}$ | $12_{\pm 17}$ | $15_{\pm 17}$ | $29_{\pm 21}$ | $1_{\pm 1}$ | $1_{\pm 2}$ | $0_{\pm 0}$ | $3_{\pm 2}$ | $53_{\pm 4}$ |
| scene-play-singletask-v0 | task1 | $94_{\pm 3}$ | $95_{\pm 2}$ | $100_{\pm 0}$ | $100_{\pm 0}$ | $30_{\pm 38}$ | $37_{\pm 16}$ | $0_{\pm 0}$ | $15_{\pm 15}$ | $100_{\pm 0}$ |
| | task2 | $12_{\pm 3}$ | $50_{\pm 13}$ | $33_{\pm 14}$ | $76_{\pm 9}$ | $2_{\pm 3}$ | $5_{\pm 10}$ | $0_{\pm 0}$ | $3_{\pm 4}$ | $100_{\pm 0}$ |
| | task3 | $32_{\pm 7}$ | $55_{\pm 16}$ | $94_{\pm 4}$ | $98_{\pm 1}$ | $0_{\pm 0}$ | $0_{\pm 0}$ | $0_{\pm 0}$ | $0_{\pm 0}$ | $95_{\pm 10}$ |
| | task4 | $0_{\pm 1}$ | $3_{\pm 3}$ | $4_{\pm 3}$ | $5_{\pm 1}$ | $0_{\pm 0}$ | $0_{\pm 0}$ | $0_{\pm 0}$ | $0_{\pm 0}$ | $95_{\pm 6}$ |
| | task5 | $0_{\pm 0}$ | $0_{\pm 0}$ | $0_{\pm 0}$ | $0_{\pm 0}$ | $0_{\pm 0}$ | $0_{\pm 0}$ | $0_{\pm 0}$ | $2_{\pm 3}$ | $93_{\pm 8}$ |
| | overall | $28_{\pm 36}$ | $41_{\pm 37}$ | $46_{\pm 44}$ | $56_{\pm 45}$ | $6_{\pm 8}$ | $8_{\pm 4}$ | $0_{\pm 0}$ | $4_{\pm 4}$ | $97_{\pm 4}$ |
| puzzle-3x3-play-singletask-v0 | task1 | $33_{\pm 6}$ | $97_{\pm 4}$ | $52_{\pm 12}$ | $90_{\pm 4}$ | $100_{\pm 0}$ | $60_{\pm 47}$ | $52_{\pm 36}$ | $5_{\pm 3}$ | $100_{\pm 0}$ |
| | task2 | $4_{\pm 3}$ | $1_{\pm 1}$ | $0_{\pm 1}$ | $16_{\pm 5}$ | $0_{\pm 0}$ | $0_{\pm 0}$ | $0_{\pm 0}$ | $0_{\pm 0}$ | $0_{\pm 0}$ |
| | task3 | $3_{\pm 2}$ | $3_{\pm 1}$ | $0_{\pm 0}$ | $10_{\pm 3}$ | $0_{\pm 0}$ | $0_{\pm 0}$ | $0_{\pm 0}$ | $0_{\pm 0}$ | $0_{\pm 0}$ |
| | task4 | $2_{\pm 1}$ | $2_{\pm 1}$ | $0_{\pm 0}$ | $16_{\pm 5}$ | $0_{\pm 0}$ | $0_{\pm 0}$ | $0_{\pm 0}$ | $0_{\pm 0}$ | $0_{\pm 0}$ |
| | task5 | $3_{\pm 2}$ | $5_{\pm 3}$ | $0_{\pm 0}$ | $16_{\pm 3}$ | $0_{\pm 0}$ | $0_{\pm 0}$ | $0_{\pm 0}$ | $2_{\pm 3}$ | $0_{\pm 0}$ |
| | overall | $9_{\pm 13}$ | $21_{\pm 38}$ | $10_{\pm 21}$ | $30_{\pm 31}$ | $20_{\pm 0}$ | $12_{\pm 9}$ | $10_{\pm 7}$ | $1_{\pm 1}$ | $20_{\pm 0}$ |
| puzzle-4x4-play-singletask-v0 | task1 | $12_{\pm 2}$ | $26_{\pm 4}$ | $48_{\pm 5}$ | $34_{\pm 8}$ | $0_{\pm 0}$ | $0_{\pm 0}$ | $0_{\pm 0}$ | $0_{\pm 0}$ | $98_{\pm 3}$ |
| | task2 | $7_{\pm 4}$ | $12_{\pm 4}$ | $14_{\pm 5}$ | $16_{\pm 5}$ | $0_{\pm 0}$ | $0_{\pm 0}$ | $0_{\pm 0}$ | $0_{\pm 0}$ | $33_{\pm 27}$ |
| | task3 | $9_{\pm 3}$ | $15_{\pm 3}$ | $34_{\pm 5}$ | $18_{\pm 5}$ | $0_{\pm 0}$ | $0_{\pm 0}$ | $0_{\pm 0}$ | $0_{\pm 0}$ | $100_{\pm 0}$ |
| | task4 | $5_{\pm 2}$ | $10_{\pm 3}$ | $26_{\pm 6}$ | $11_{\pm 3}$ | $0_{\pm 0}$ | $0_{\pm 0}$ | $0_{\pm 0}$ | $0_{\pm 0}$ | $85_{\pm 14}$ |
| | task5 | $4_{\pm 1}$ | $7_{\pm 3}$ | $24_{\pm 11}$ | $7_{\pm 3}$ | $0_{\pm 0}$ | $0_{\pm 0}$ | $0_{\pm 0}$ | $0_{\pm 0}$ | $72_{\pm 40}$ |
| | overall | $7_{\pm 4}$ | $14_{\pm 8}$ | $29_{\pm 13}$ | $17_{\pm 10}$ | $0_{\pm 0}$ | $0_{\pm 0}$ | $0_{\pm 0}$ | $0_{\pm 0}$ | $78_{\pm 13}$ |

## C  D4RL RESULTS

We report the scores for the D4RL (Fu et al., 2020) environments, which has been used as a standard dataset for offline RL evaluation. Same as OGBench experiments, we report the normalized score across 4 seeds and report the standard deviation for each tasks. The results for prior works are reported following their respective papers. **MOPO**$^*$ is an improved version of **MOPO**, introduced in Sun et al. (2023). We highlight the numbers that are above or equal to $95\%$ of the best performance.

For sequence modeling methods, we consider **TT** (Janner et al., 2021a), which predicts offline trajectory with a Transformer model (Vaswani et al., 2017) and find the best trajectory by conditioning with target return-to-go, and **TAP** (Jiang et al., 2023), which improves TT by quantizing the action space with VQ-VAE (van den Oord et al., 2017).

Table 14: **D4RL MuJoCo Gym results.**

| Dataset | Model-free | | | Seq. modeling | | Model-based | | | |
|---|---|---|---|---|---|---|---|---|---|
| | CQL | ReBRAC | IQL | TT | TAP | MOPO$^*$ | MOBILE | LEQ | MAC |
| hopper-r | 5 | 8 | 7 | 6 | - | 31 | 32 | 32 | $28_{\pm 3}$ |
| hopper-m | 61 | 102 | 66 | 67 | 63 | 62 | 102 | 103 | $92_{\pm 4}$ |
| hopper-mr | 86 | 98 | 94 | 99 | 87 | 99 | 104 | 103 | $95_{\pm 2}$ |
| hopper-me | 96 | 107 | 91 | 110 | 105 | 81 | 111 | 109 | $110_{\pm 1}$ |
| walker2d-r | 5 | 18 | 5 | 5 | - | 7 | 17 | 21 | $5_{\pm 1}$ |
| walker2d-m | 79 | 82 | 78 | 84 | 64 | 81 | 87 | 74 | $82_{\pm 3}$ |
| walker2d-mr | 76 | 77 | 73 | 89 | 66 | 85 | 92 | 98 | $86_{\pm 6}$ |
| walker2d-me | 109 | 112 | 109 | 101 | 107 | 112 | 117 | 108 | $108_{\pm 1}$ |
| halfcheetah-r | 31 | 30 | 11 | 6 | - | 38 | 32 | 30 | $12_{\pm 0}$ |
| halfcheetah-m | 46 | 66 | 47 | 46 | 45 | 73 | 74 | 71 | $47_{\pm 1}$ |
| halfcheetah-mr | 45 | 51 | 44 | 44 | 40 | 72 | 66 | 65 | $38_{\pm 1}$ |
| halfcheetah-me | 95 | 101 | 86 | 95 | 91 | 90 | 105 | 102 | $68_{\pm 2}$ |
| Total | 740 | 852 | 717 | 747 | - | 844 | 960 | 923 | 771 |

# D   TRAINING CURVES

We provide the training curve of MAC for large-scale, long-horizon environments and reward-based environment in Figure 5 and Figure 6 (corresponding to Table 1 and Table 2). We plot the mean and the standard deviation (across 4 seeds) by covering [mean - std, mean + std] area with a lighter color.

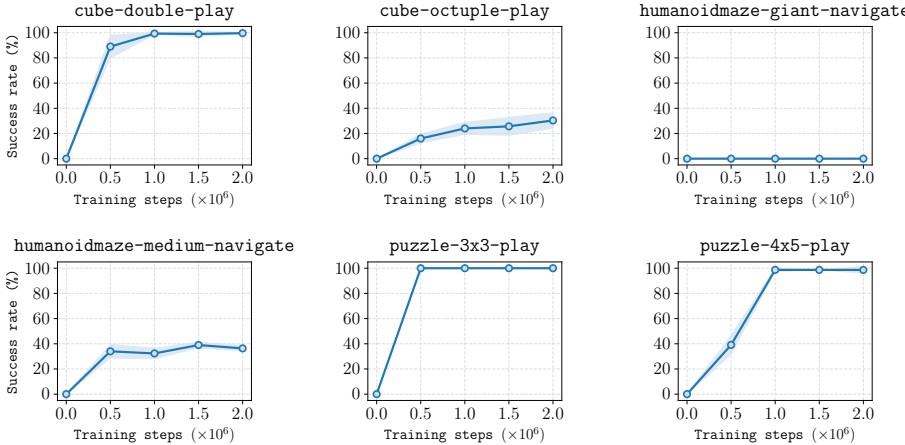

Figure 5: **Training curve of MAC in large-scale, long-horizon environments.** We report the success rate for 15 evaluation episodes across 4 seeds (total 60 episodes). Shaded region represents the [mean - std, mean + std].

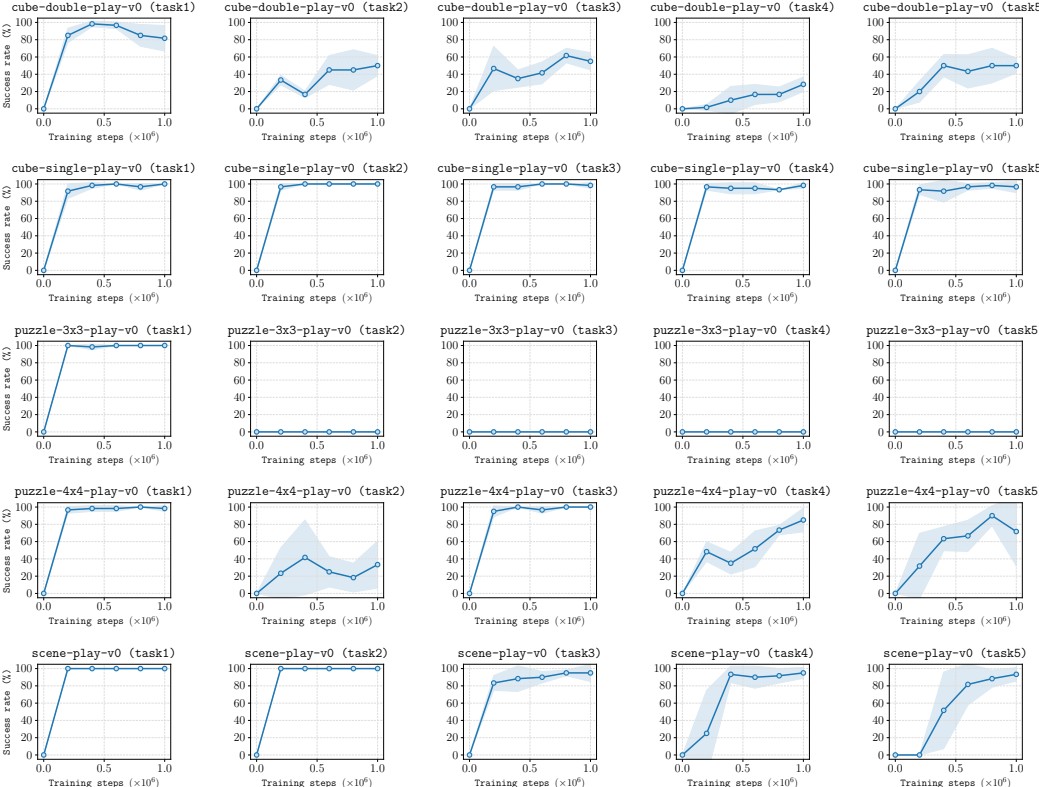

Figure 6: **Training curve of MAC in reward-based environments.** We report the success rate for 15 evaluation episodes across 4 seeds (total 60 episodes). Shaded region represents the [mean - std, mean + std].

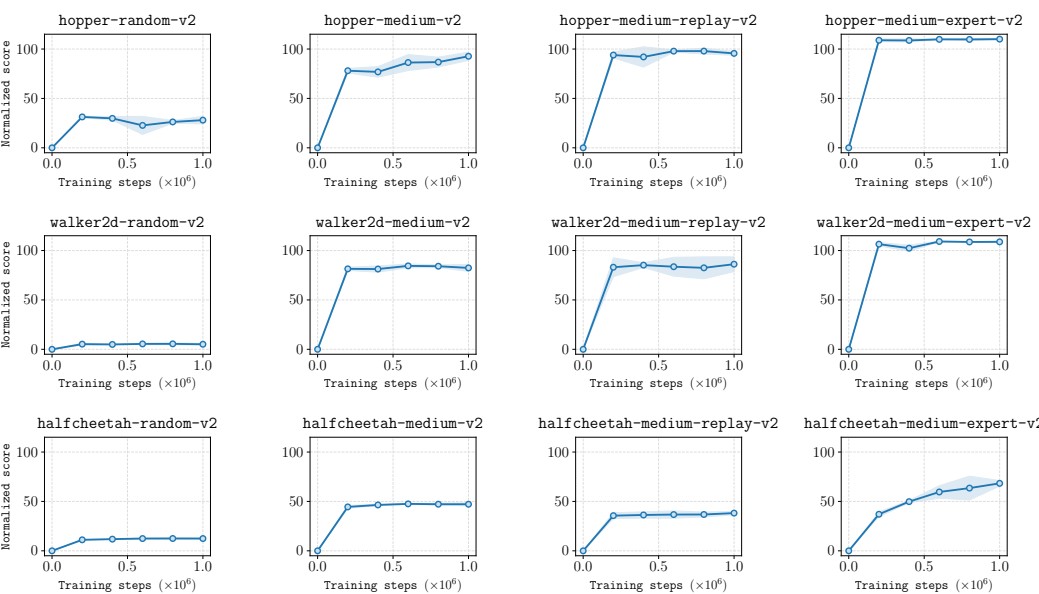

Figure 7: **Training curve of MAC in D4RL environments.** We report the success rate for 15 evaluation episodes across 4 seeds (total 60 episodes). Shaded region represents the [mean - std, mean + std].

# E    MORE ABLATION EXPERIMENTS

**Q: Is distillation with $\pi_\theta$ necessary?**

**A:** To understand the importance of this component, we conduct an ablation study of MAC removing the distillation. Specifically, we directly train the one-step flow model $\pi_\omega$ with flow matching BC loss, instead of distilling the multi-step flow model $\pi_\theta$. We present the ablation results on the default tasks for five reward-based environments used in Table 15. The results indicate that the use of $\pi_\theta$ is crucial for MAC in OGBench tasks where behavioral policies are highly multi-modal.

Table 15: **Ablation of using $\pi_\theta$.**

| Task | MAC (w/o $\pi_\theta$) | MAC |
|------|------|------|
| cube-single-play-v0 | $2_{\pm3}$ | $100_{\pm0}$ |
| cube-double-play-v0 | $0_{\pm0}$ | $50_{\pm12}$ |
| scene-play-v0 | $5_{\pm9}$ | $100_{\pm0}$ |
| puzzle-3x3-play-v0 | $2_{\pm3}$ | $0_{\pm0}$ |
| puzzle-4x4-play-v0 | $15_{\pm6}$ | $85_{\pm14}$ |

**Q: How does model error correlate with the performance?**

**A:** Figure 8 shows the policy performance and rollout error with respect to the rollout length for various action chunk sizes. For chunk sizes of 1, 5, and 10, we observe a consistent trend that at a given rollout length, smaller chunks produce larger model-prediction errors and correspondingly lower policy performance (clearly shown in rollout length of 50 and 100). However, excessively large chunk sizes (25) breaks this trend, where it achieves lower rollout error, but

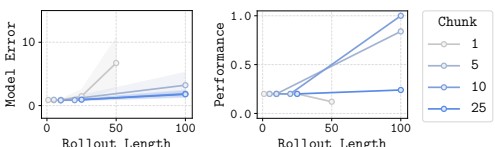

Figure 8: **Model error correlate with the performance, unless action chunk size is too large.**

does not yield better performance. It is because while larger chunks helps reducing the compounding model error, they also make both policy learning and Q-function estimation harder since the action space grows exponentially. While flow rejection sampling mitigates this problem by limiting the action sequence to in-distribution actions, extremely large chunk sizes exacerbate this problem, hurting the performance despite improved model accuracy.

