# OpenReview forum: "Scalable Offline Model-Based RL with Action Chunks"
_ICLR.cc/2026/Conference — ICLR 2026 Poster_

### Official Review · Reviewer_P1Zz · 2025-10-18

**Soundness:** 3
**Presentation:** 3
**Contribution:** 2
**Rating:** 4
**Confidence:** 5

**Summary:**

The paper introduces Model-Based Reinforcement Learning with Action Chunks (MAC), a scalable offline model-based RL framework designed to handle long-horizon tasks. The key idea is to mitigate the trade-off between long-horizon bias reduction and model error accumulation in model-based value expansion by using action-chunk models—multi-step dynamics that predict future states from sequences of actions rather than single actions. Complementarily, an action-chunk policy is trained via flow matching and rejection sampling to model complex, high-dimensional action distributions while avoiding out-of-distribution exploitation. Experiments across large-scale datasets (100M transitions) and standard offline RL benchmarks show that MAC achieves state-of-the-art results among model-based methods and competitive performance compared to model-free baselines, particularly in complex, long-horizon manipulation tasks (Tables 1–2; Sec. 5.1–5.2).

**Strengths:**

- This paper focuses on model-based value expansion in the offline setting, aiming to balance value bias reduction and model error reduction. This is a highly valuable and meaningful problem, as it contributes to improving the performance of offline reinforcement learning on long-horizon tasks.
- The method proposed in the paper mainly adopts a multi-step dynamics model (action-chunk model) to reduce rollout errors and uses rejection sampling to model complex, high-dimensional action distributions. Such an approach is intuitively effective, and the experiments show that MAC achieves outstanding performance on long-horizon offline RL tasks.

**Weaknesses:**

- The multi-step dynamics model (action-chunk model), which takes s_t and a multi-step action sequence (a_t, a_{t+1}, …, a_{t+k-1}) as input to predict s_{t+k}, has already been proposed in previous works [1][2][3]. The multi-step dynamics model used in MAC does not differ from these earlier approaches. Similarly, the flow policy [4], model-based value expansion [5] are also not introduced for the first time in this paper. Therefore, I believe the contribution of this work is relatively limited.
- The application of multi-step dynamics models to the offline RL domain has been explored in ADMPO [6]. Similar to MAC, ADMPO investigates how multi-step dynamics models can reduce rollout errors in the offline setting. The key difference is that ADMPO adopts a more traditional policy learning approach, representing another way of leveraging multi-step dynamics models in offline settings. I believe ADMPO is highly related to MAC, as ADMPO has already demonstrated that multi-step dynamics models can significantly improve policy optimization performance in offline RL. Therefore, MAC should be compared against ADMPO.
- Although this paper targets long-horizon offline tasks, it is inappropriate for an offline RL algorithm to lack experiments on the standard D4RL benchmark.


[1] Kavosh Asadi, Evan Cater, Dipendra Misra, and Michael L. Littman. Towards a simple approach to
multi-step model-based reinforcement learning. CoRR, abs/1811.00128, 2018.
[2] Kavosh Asadi, Dipendra Misra, Seungchan Kim, and Michael L. Littman. Combating the
compounding-error problem with a multi-step model. CoRR, abs/1905.13320, 2019.
[3] Tong Che, Yuchen Lu, George Tucker, Surya Bhupatiraju, Shane Gu, Sergey Levine, and Yoshua
Bengio. Combining model-based and model-free RL via multi-step control variates. https://
openreview.net, 2018. URL https://openreview.net/forum?id=HkPCrEZ0Z.
[4] Seohong Park, Qiyang Li, and Sergey Levine. Flow q-learning. In International Conference on
Machine Learning (ICML), 2025c.
[5] Vladimir Feinberg, Alvin Wan, Ion Stoica, Michael I. Jordan, Joseph E.
Gonzalez, and Sergey Levine, ‘Model-based value estimation for ef-
ficient model-free reinforcement learning’, CoRR, abs/1803.00101,
(2018).
[6] Haoxin Lin, Yu-Yan Xu, Yihao Sun, Zhilong Zhang, Yi-Chen Li, Chengxing Jia, Junyin Ye, Jiaji
Zhang, and Yang Yu. Any-step dynamics model improves future predictions for online and
offline reinforcement learning. In ICLR 2025.

**Questions:**

- Could the authors include a performance comparison with ADMPO? Since both methods apply multi-step dynamics models to offline RL, such a comparison would better highlight whether MAC offers advantages in terms of methodological design.
- Could the authors provide experimental results on D4RL? This is essential for evaluating the performance of an offline RL algorithm.
- The experiments in the paper show that the multi-step dynamics model effectively reduces rollout errors, and in particular, the larger the chunk size, the more significant the reduction—this aligns well with intuition. However, it remains unclear whether MAC’s performance gain truly comes from the reduction in rollout error, or merely from the flow action-chunk policy. Although the paper provides results for different chunk sizes, this does not fully address the issue. Could the authors provide additional experiments comparing MAC’s performance across different chunk sizes and rollout lengths, explicitly reporting the rollout error and policy performance for each configuration? This would help verify whether smaller rollout errors indeed correlate with better policy performance.
- From the experiments in the paper, when the action-chunk length is set to 25 the rollout error is still effectively reduced, but the algorithm’s performance degrades. Could the authors provide a deeper explanation for this?

---

> ### Author Response · Authors · 2025-11-25
> **Response to Reviewer P1Zz (1/2)**
>
> Thank you for your constructive and detailed feedback. We address your questions and concerns in detail below and have revised the paper accordingly (please refer to the updated PDF). Specifically, we believe the added baseline and ablation experiments have strengthened the paper. Please let us know if you have any additional concerns or questions. If we have fully addressed your concerns, we would appreciate it if you could update the score accordingly.
>
> &nbsp;
>
>
> **W1. The multi-step dynamics model has already been proposed in previous works [1][2][3]. Similarly, the flow policy [4], model-based value expansion [5] are also not introduced for the first time in this paper. Therefore, I believe the contribution of this work is relatively limited.**
>
> While multi-step models and flow policies have appeared individually, MAC’s contribution is in integrating action-chunk models, expressive flow behavior policies, and value-guided rejection sampling into a single offline MBRL framework that vastly reduces bias of value learning with **stable 100-step on-policy rollouts**, which has never been achieved by prior offline MBRL approaches.
>
> To our knowledge, no prior offline MBRL approach achieves long-horizon rollout stability at this scale. Even “Any-step dynamics” [1] uses task-dependent rollout lengths (often 5–20, rarely 50), whereas MAC consistently supports long rollouts across tasks, as we also strongly restrict the actions to be in-distribution via rejection sampling, so that the input of the model is more likely to stay in-distribution during rollouts.
>
> &nbsp;
>
>
> **W2 & Q1. Could the authors include a performance comparison with ADMPO? Since both methods apply multi-step dynamics models to offline RL, such a comparison would better highlight whether MAC offers advantages in terms of methodological design.**
>
> We run ADMPO using their official codebase and provide the result across 4 seeds for task1 variants of OGBench singletask benchmarks. As shown in the table below, MAC consistently outperforms ADMPO across all tasks.
>
> | task | MAC (ours) | ADMPO |
> |---|---|---|
> | cube-single-singletask-task1 | $100.0 \pm 0.0 $ | $87.6 \pm 2.5$ |
> | cube-double-singletask-task1 | $81.7 \pm 14.8 $ | $0.1 \pm 0.1$ |
> | puzzle-3x3-singletask-task1 | $100.0 \pm 0.0 $ | $0.1 \pm 0.1$ |
> | puzzle-4x4-singletask-task1 | $98.3 \pm 3.3 $ | $0.0 \pm 0.0$ |
> | scene-singletask-task1 | $100.0 \pm 0.0 $ | $0.0 \pm 0.0$ |
>
> While we are continuing to run the full benchmark, as ADMPO requires roughly 36 hours per task on our hardware (A5000), and we are searching from 16 hyperparameter settings ($m = 2, H \in \{5, 10, 25, 50\}, \beta \in \{1, 5, 10, 20\}) per task, it is unlikely to complete all runs within the rebuttal period. We will update the table in the final version as full results become available.
>
> &nbsp;
>
>
> **W3 & Q2. Could the authors provide experimental results on D4RL? This is essential for evaluating the performance of an offline RL algorithm.**
>
> We want to clarify that our focus is on long-horizon, large-scale tasks (OGBench), which better stress-test horizon scalability. These tasks highlight MAC’s core strength: reducing TD bias via stable 100-step rollouts. In contrast, D4RL locomotion tasks are short-horizon and dense-reward, where long-horizon rollouts offer limited benefit, as also noted in prior works [2, 3, 4].
>
> Nevertheless, we have added D4RL results in Table 14 (Section C) and its training curve in Figure 7 (Section D). MAC matches or surpasses the performance of prior model-based methods on Hopper and Walker2D, but underperforms on HalfCheetah and on random datasets. We found that MAC works relatively worse in random datasets because the flow behavior policy collapses to a random policy, which makes flow-based rejection sampling far less effective. Moreover, the action-chunked flow policy is hard to model reactive policies, which is important in D4RL locomotion tasks (we believe this is also related to the relatively low performance in humanoidmaze tasks in OGBench experiments). We have added a discussion about these limitations in Section 6.
>
> &nbsp;

---

> > ### Author Response · Authors · 2025-11-25
> > **Response to Reviewer P1Zz (2/2)**
> >
> > **Q3. Could the authors provide additional experiments comparing MAC’s performance across different chunk sizes and rollout lengths, explicitly reporting the rollout error and policy performance for each configuration? This would help verify whether smaller rollout errors indeed correlate with better policy performance.**
> >
> > Thank you for the suggestion. We added the graph in Figure 8 (Section E) that reports both policy performance and rollout error for each configuration. For chunk sizes of 1, 5, and 10, we observe a consistent trend that at a given rollout length, smaller chunks produce larger model-prediction errors and correspondingly lower policy performance (clearly shown in rollout lengths of 50 and 100).
> >
> > However, excessively large chunk sizes (25) break this trend; i.e., having smaller rollout errors alone does not lead to better policy performance. We explain this behavior in the following response for Q4.
> >
> > &nbsp;
> >
> >
> > **Q4. From the experiments in the paper, when the action-chunk length is set to 25 the rollout error is still effectively reduced, but the algorithm’s performance degrades. Could the authors provide a deeper explanation for this?**
> >
> > Larger chunks reduce compounding model error but make both policy learning and Q-function estimation harder as the action space grows exponentially. While flow rejection sampling mitigates this problem by limiting the action sequence to in-distribution actions, extremely large chunk sizes exacerbate this problem, degrading the performance despite improved model accuracy. We added the discussion of this point in Section 6 (L498).
> >
> > &nbsp;
> >
> > **References**
> >
> > [1] Haoxin Lin et al., Any-step Dynamics Model Improves Future Predictions for Online and Offline Reinforcement Learning, ICLR 2025.
> >
> > [2] Siddarth Venkatraman et al., Reasoning with Latent Diffusion in Offline Reinforcement Learning, ICLR 2024.
> >
> > [3] Zibin Dong et al., DiffuserLite: Towards Real-time Diffusion Planning, NeurIPS 2024.
> >
> > [4] Kwanyoung Park et al., Model-based Offline Reinforcement Learning with Lower Expectile Q-Learning, ICLR 2025.

---

> > > ### Comment · Reviewer_P1Zz · 2025-11-25
> > > **Thank you**
> > >
> > > Thank you for your response. I believe the newly added experiments have effectively strengthened the paper.
> > >
> > > Judging from the various experimental results, I attribute the success of MAC primarily to the reduction of roll-out errors via multi-step prediction and the prevention of OOD (out-of-distribution) issues through constraints on the roll-out policy distribution.
> > >
> > > MAC demonstrates strong performance on truly long-horizon tasks and proves to be a highly effective method. I believe its publication will be of great benefit to the model-based RL community.
> > >
> > > I have no further questions or concerns. Given that the authors have thoroughly addressed all my comments, I am pleased to raise my score from 4 to 8.

---

### Official Review · Reviewer_ZGYu · 2025-10-26

**Soundness:** 3
**Presentation:** 3
**Contribution:** 3
**Rating:** 6
**Confidence:** 4

**Summary:**

Model-based offline reinforcement learning (RL) approaches often face a key dilemma: model rollouts must be sufficiently long for the value function to capture accurate long-horizon information, yet they suffer from error accumulation as the rollout horizon increases. To address this, the paper employs action chunks, a common technique in robotics that predicts a sequence of actions rather than a single step. This reduces the number of function calls needed to generate future trajectories, enabling longer horizons while simultaneously mitigating error accumulation. Once the value functions are obtained, the policy is implicitly extracted via rejection sampling from an expressive behavior policy modeled by a distilled one-step flow matching policy.  Experiments on OGBench and D4RL demonstrate the superior performance of the proposed method, MAC (Model-based Offline RL with Action Chunks), in robotic manipulation tasks. Nonetheless, the method falls short in locomotion tasks that require precise control, where approaches based on differentiable reward maximization or policy gradients may be more effective. Moreover, all experiments are conducted with low-dimensional state inputs, without evaluation on image-based tasks.

**Strengths:**

1. Scale up RL methods to tackle long-horizon problems is very important.
2. The idea of adopting action chunks to address the dilemma between long horizons and small functional calls is quite straightforward.
3. The proposed method (MAC) is quite simple to be implemented. So, this paper offers a good starting point for future research on this direction.
4. The empirical performance on robotics manipulation tasks in OGBench are also strong, offering potential solutions for robotics.

**Weaknesses:**

I have a few concerns regarding the novelty and the complexity of the proposed method:

1. `Novelty of Action Chunks` The idea of adopting action chunks to extend RL horizons is not particularly novel. In fact, a recent work [1] has already explored a model-free version of this idea. Therefore, the novelty contribution of this paper is somewhat limited. However, since action chunking appears to be a principled and simple approach to addressing the dilemma highlighted by the authors, I would not consider the lack of novelty a major concern.

2. `Method Complexity` While the high-level idea of MAC is straightforward and the implementation relatively simple and robust to hyperparameter variations, the method introduces multiple components, including a flow policy, a distilled flow policy, a reward function, a state value function, a state-action value function, and a dynamics model. This design may help address the horizon-scaling issue, but it also adds significant complexity, potentially restricting MAC to smaller model sizes. In the long run, a unified model or framework that consolidates these components would be preferable. Nonetheless, given that the primary contribution of MAC lies in leveraging action chunks to scale up model-based RL horizons, I would also not consider this limitation a major concern.

[1] Reinforcement Learning with Action Chunking, 2025.

**Questions:**

1. I understand that rejection sampling from a behavior policy implicitly imposes a behavior constraint. However, I still wonder whether additional penalties are needed to mitigate value overestimation. Can simply applying best-of-N sampling alone account for the scale of performance improvements reported?

At this point, I have no further questions. Please see my comments under Weaknesses for more details.

---

> ### Author Response · Authors · 2025-11-25
> **Response to Reviewer ZGYu**
>
> Thank you for your constructive feedback. We address your questions and concerns in detail below and have revised the paper accordingly (please refer to the updated PDF). Specifically, we believe the clarifications on the novelty and the complexity of the model have strengthened the paper. Please let us know if you have any additional concerns or questions. If we have fully addressed your concerns, we would appreciate it if you could update the score accordingly.
>
>
> &nbsp;
>
> **W1. Novelty of Action Chunks**
>
>
> While action-chunking has already been explored in RL, MAC’s contribution is to scale offline MBRL into long-horizon tasks by vastly reducing bias of value learning with **stable 100-step rollouts**, which has never been achieved by prior offline MBRL approaches, by integrating action chunking with expressive flow behavior policies, and value-guided rejection sampling.
>
> To our knowledge, no prior offline MBRL approach achieves long-horizon rollout stability at this scale. Even “Any-step dynamics” [1] uses task-dependent rollout lengths (often 5–20, rarely 50), whereas MAC consistently supports long rollouts across tasks.
>
> &nbsp;
>
> **W2. Method Complexity**
>
> We would like to clarify that MBRL generally requires a larger number of components than model-free RL. For example, prior works such as MOBILE, MOPO, and LEQ require a transition model, reward model, Q function, and policy. Compared to those methods, the **only** additional component of MAC is the one-step distilled flow policy (modulo the difference between Q and V). Moreover, we note that training this additional distillation policy does not require any hyperparameters.
>
> Furthermore, MAC requires even fewer per-task hyperparameters, thanks to the stability of rejection sampling. For example, MOPO, MOBILE, and LEQ require $2$ per-task hyperparameters (horizon length, penalization coefficient), whereas our method does not require any. Hence, we believe MAC is even simpler to use than previous MBRL approaches in practice, despite its ostensible complexity.
>
>
> &nbsp;
>
> **Q1.  I understand that rejection sampling from a behavior policy implicitly imposes a behavior constraint. However, I still wonder whether additional penalties are needed to mitigate value overestimation. Can simply applying best-of-N sampling alone account for the scale of performance improvements reported?**
>
> Thank you for sharing your thoughts on using rejection sampling to address the value overestimation problem. Since overestimation arises from sampling out-of-distribution state-action pairs, restricting sampling to pairs in the dataset leads to an unbiased value estimate. In other words, if the world model and the flow policy perfectly models the distribution in the training set (in deterministic environments), then model rollouts are unbiased, obviating the need of additional penalization.
> While this setting is admittedly idealized (in practice, world model errors are non-zero (Figure 2) and the flow policy may still propose out-of-distribution actions), we found that explicit regularization was not required in our experiments. That said, we agree that this discrepancy remains as a potential concern, and further investigation in this issue will be an important future work.
>
> &nbsp;
>
> **References**
>
> [1] Haoxin Lin et al., Any-step Dynamics Model Improves Future Predictions for Online and Offline Reinforcement Learning, ICLR 2025.

---

> > ### Comment · Reviewer_ZGYu · 2025-11-25
> > **Thanks for the detailed rebuttal**
> >
> > Thanks for the rebuttal. I have no additional concerns and would maintain the positive recommendation for this paper.

---

### Official Review · Reviewer_Qjkz · 2025-10-29

**Soundness:** 3
**Presentation:** 2
**Contribution:** 2
**Rating:** 4
**Confidence:** 4

**Summary:**

This paper introduces MAC, an offline MBRL algorithm designed particularly for long-horizon tasks. They use action-chunking to reduce the number of recursive calls to the dynamics model to hopefully reduce compounding error. They also use flow-based rejection sampling to aim to prevent model exploitation.

**Strengths:**

The issue they address is relevant, the results seem strong, and the paper is reasonably clear in explaining a complicated method.

**Weaknesses:**

**W1.** There are so many moving parts, including training 6 models. This is expensive, and probably also more difficult to implement and debug compared to many other methods.

**W2.** The main benchmark in offline RL is probably D4RL. The authors have not provided results on this.

**W3.** As far as I understand the proposed method is roughly N times more expensive than prior algorithms(?) Furthermore, N is high (~32), and they potentially need larger models since action-chunk inputs to the model are n times larger. The authors should include a comparison and discussion of the runtime to help justify this significant extra cost.

**W4.** Performance appears highly sensitive to the action-chunk length hyperparameter. This could make it difficult to tune, especially offline.

More weaknesses: See Questions below.

**Questions:**

**Q1.** What do the authors mean by “Thanks to the expressivity of the flow BC policy, we query the model only with in-distribution
action-chunk samples”? Relatedly, isn't it easier to be OOD since "action" space is exponentially larger in the number of action steps included as input to the model?

**Q2.** The baselines are a mixture of results from the prior papers and from the author's reimplementations. Are the authors confident they understood implementation details that might have unfairly weakened results from prior paper, and also that they correctly implemented and fairly tuned their reimplemented baselines?

**Q3.**  Why was flow matching chosen over diffusion? Did the authors try/consider diffusion as an alternative?

**Q4.** The method involves training 6 models at the same time (?). What was this like in terms of convergence and stability? It would be interesting to include training curves.

**Q5.** Related to ^, often you would train the dynamics model p_\phi, r_\phi to convergence first, and then train the policy. Why train them side by side?

**Q6.** Is \pi_\theta really needed if you then replace it with a less expressive model. For completeness, the authors should include and ablation bypassing \pi_\theta to show whether it is important to their method.

**Q7.** (Comment) “one-step” is confusing since "steps" are also used to refer to RL environment steps. Is there another term you can use?

---

> ### Author Response · Authors · 2025-11-25
> **Response to Reviewer Qjkz (1/3)**
>
> Thank you for your keen and detailed feedback. We address your questions and concerns in detail below and have revised the paper accordingly (please refer to the updated PDF). Specifically, we believe the added experiments (D4RL and bypassing $\pi_\theta$) and clarifications (training time and model complexity) have strengthened the paper. Please let us know if you have any additional concerns or questions. If we have fully addressed your concerns, we would appreciate it if you could update the score accordingly.
>
> &nbsp;
>
> **W1. There are so many moving parts, including training 6 models. This is expensive, and probably also more difficult to implement and debug compared to many other methods.**
>
> We would like to clarify that MBRL generally requires a larger number of components than model-free RL. For example, prior works such as MOBILE, MOPO, and LEQ require a transition model, reward model, Q function, and policy. Compared to those methods, the **only** additional component of MAC is the one-step distilled flow policy (modulo the difference between Q and V). Moreover, we note that training this additional distillation policy does not require any hyperparameters.
>
> Furthermore, MAC requires even fewer per-task hyperparameters, thanks to the stability of rejection sampling. For example, MOPO, MOBILE, and LEQ require $2$ per-task hyperparameters (horizon length, penalization coefficient), whereas our method does not require any. Hence, we believe MAC is even easier to debug than previous MBRL approaches in practice, despite its ostensible complexity.
>
> &nbsp;
>
> **W2. The main benchmark in offline RL is probably D4RL. The authors have not provided results on this.**
>
> We want to clarify that our focus is on long-horizon, large-scale tasks (OGBench), which better stress-test horizon scalability. These tasks highlight MAC’s core strength: reducing TD bias via stable 100-step rollouts. In contrast, D4RL locomotion tasks are short-horizon and dense-reward, where long-horizon rollouts offer limited benefit, as also noted in prior works [1, 2, 3].
>
> Nevertheless, we have added D4RL results in Table 14 (Section C) and its training curve in Figure 7 (Section D). MAC matches or surpasses the performance of prior model-based methods on Hopper and Walker2D, but underperforms on HalfCheetah and on random datasets. We found that MAC works relatively worse in random datasets because the flow behavior policy collapses to a random policy, which makes flow-based rejection sampling far less effective. Moreover, the action-chunked flow policy is hard to model reactive policies, which is important in D4RL locomotion tasks (we believe this is also related to the relatively low performance in humanoidmaze tasks in OGBench experiments). We have added a discussion about these limitations in Section 6.
>
> &nbsp;
>
> **W3. As far as I understand the proposed method is roughly N times more expensive than prior algorithms(?) Furthermore, N is high (~32), and they potentially need larger models since action-chunk inputs to the model are n times larger. The authors should include a comparison and discussion of the runtime to help justify this significant extra cost.**
>
>
> We would like to clarify that our method is **not** necessarily “N times slower” during training because we do not need to backpropagate the gradient for rejection sampling, where learning the parameters using the backpropagation (e.g., value learning) mostly consists of the cost. We added the clarification on this issue in Section 4.2 (L319).
>
> To quantitatively support this claim, we added the table for average training times on an A5000 GPU for both single-task and multi-task settings across MAC and prior MBRL baselines in Table 4 (Section A.2). MAC trains in 3.1 hours on average for a single task, while prior methods take 1.4 ~ 2.6 hours to train. All models use identical architecture sizes across methods.
>
> Moreover, we added the table for average inference time in the same settings in Table 5 (Section A.2). We did not observe meaningful inference slowdowns on OGBench, as rejection sampling is parallelizable. That being said, rejection sampling may potentially be a bottleneck in settings where inference speed cannot benefit from parallelization (e.g., using very large VLA-style models). Real-world deployment of MAC would be a promising future direction.

---

> ### Author Response · Authors · 2025-11-25
> **Response to Reviewer Qjkz (2/3)**
>
> **W4. Performance appears highly sensitive to the action-chunk length hyperparameter. This could make it difficult to tune, especially offline.**
>
> We appreciate the reviewer pointing out the sensitivity to the action-chunk length. As the reviewer mentioned, we did observe that certain chunk lengths (e.g., 1, 5, 25) were suboptimal, while a moderate value (10) consistently worked well. However, once identified, this setting **transferred cleanly across all tasks**. We used the same rollout length (=10) and chunk size (=10) everywhere without further tuning. In this sense, MAC was **easier to tune** than prior model-based baselines such as MOBILE and MOPO, which require task-specific rollout horizons and uncertainty penalties to remain stable. We have clarified this point in the revised PDF in Section 4.2 (L330).
>
> &nbsp;
>
> **Q1. What do the authors mean by “Thanks to the expressivity of the flow BC policy, we query the model only with in-distribution action-chunk samples”? Relatedly, isn't it easier to be OOD since "action" space is exponentially larger in the number of action steps included as input to the model?**
>
> Thank you for pointing out the ambiguity. It is true that action space grows exponentially, making the behavior policy learning harder. In this context, we meant that **compared to Gaussian policies**, the flow-based behavior policy better models the distribution of actions in the data, such that action chunks sampled from the model are more likely to actually be in-distribution. That is, while the combinatorial action space indeed grows exponentially, the flow model’s expressivity helps maintain coverage over likely trajectories and reduces OOD queries to the dynamics model. We have clarified this point in the updated PDF (Section 4.1, L245).
>
> &nbsp;
>
> **Q2. The baselines are a mixture of results from the prior papers and from the author's reimplementations. Are the authors confident they understood implementation details that might have unfairly weakened results from prior paper, and also that they correctly implemented and fairly tuned their reimplemented baselines?**
>
> To address the concern in the experimental setting for the baseline, we run the official codebase of MOBILE in two reward-based OGBench tasks (cube-single-task1, puzzle-3x3-task1) and report the result across 4 seeds in the below table. We tuned the hyperparameter in the range of $H \in \{1, 5, 10\}, \beta \in \{0.1, 0.3, 0.5, 1, 5, 10, 20\}$, which is even broader than the values suggested in the original paper. As shown in the table below, the official implementation performs worse compared to our implementation.
>
> | task | MOBILE (official) | MOBILE (ours) |
> |---|---|---|
> | cube-single-task1-v0 | $0 \pm 0$  | $85 \pm 22$ |
> | puzzle-3x3-task1-v0 | $0 \pm 0$ | $60 \pm 47$ |
>
> Moreover, to ensure transparency, we have released all baseline implementations in the supplementary materials, and the list of hyperparameters we used for tuning, for full reproducibility and external verification.
>
> &nbsp;
>
>
> **Q3. Why was flow matching chosen over diffusion? Did the authors try/consider diffusion as an alternative?**
>
> We chose flow matching because it is easier to implement and generally robust to noise schedule [4]. Nevertheless, MAC can be combined with diffusion policies, and we believe it will be an interesting direction.
>
> &nbsp;
>
> **Q4 & Q5. The method involves training 6 models at the same time (?). What was this like in terms of convergence and stability? It would be interesting to include training curves. Also, why train them side by side?**
>
> We train all 6 models jointly because we can parallelize gradient updates with JAX, greatly improving training speed. Empirically, we observed no performance difference between joint and sequential training. The training curves of MAC show that the algorithm converges stably. We included these curves in Figures 5 and 6 (Section D) in the revised pdf.

---

> ### Author Response · Authors · 2025-11-25
> **Response to Reviewer Qjkz (3/3)**
>
> **Q6. Is \pi_\theta really needed if you then replace it with a less expressive model. For completeness, the authors should include and ablation bypassing \pi_\theta to show whether it is important to their method.**
>
> Thank you for the suggestion. We run the ablation experiment on bypassing $\pi_\theta$ in the default tasks of the reward-based environments. Specifically, we directly train the one-step flow model $\pi_\omega$ with flow matching BC loss, instead of distilling the multi-step flow model (removes $\pi_\theta$). We added the experimental results and the analysis in Table 15 (Section E).
>
> | task | MAC (w/o $\pi_\theta$) | MAC |
> |---|---|---|
> | cube-single-play-v0 | $2 \pm 3 $ | $100 \pm 0$ |
> | cube-double-play-v0 | $0 \pm 0 $ | $50 \pm 12$ |
> | puzzle-3x3-play-v0 | $2 \pm 3$ | $0 \pm 0$ |
> | puzzle-4x4-play-v0 | $15 \pm 6 $ | $85 \pm 14$ |
> | scene-play-v0 | $5 \pm 9 $ | $100 \pm 0$ |
>
>
> The results show that MAC without $\pi_\theta$ largely underperforms the original MAC. It  indicates that the use of full flow policy $\pi_\theta$ is crucial for MAC in OGBench tasks, where behavioral policies are highly multi-modal.
>
> &nbsp;
>
> **Q7. (Comment) “one-step” is confusing since "steps" are also used to refer to RL environment steps. Is there another term you can use?**
>
> We appreciate the reviewer pointing out the potential ambiguity. However, “one-step” (or “single-step”) is the standard term for single-step distillation procedures (e.g., as used in [4, 5, 6]). Because this terminology is already conventional, we would like to retain “one-step” for consistency with prior work, but please let us know if you have alternative suggestions for this terminology. We have clarified this terminology in Section 4.2 (L321) of the revised paper.
>
> &nbsp;
>
> **References**
>
> [1] Siddarth Venkatraman et al., Reasoning with Latent Diffusion in Offline Reinforcement Learning, ICLR 2024.
>
> [2] Zibin Dong et al., DiffuserLite: Towards Real-time Diffusion Planning, NeurIPS 2024.
>
> [3] Kwanyoung Park et al., Model-based Offline Reinforcement Learning with Lower Expectile Q-Learning, ICLR 2025.
>
> [4] Seohong Park et al., Flow Q-Learning, ICML 2025.
>
> [5] Kevin Frans et al., One Step Diffusion via Shortcut Models, ICLR 2025.
>
> [6] Haoxin Lin et al., Any-step Dynamics Model Improves Future Predictions for Online and Offline Reinforcement Learning, ICLR 2025.

---

### Official Review · Reviewer_9Rxo · 2025-11-02

**Soundness:** 3
**Presentation:** 3
**Contribution:** 3
**Rating:** 6
**Confidence:** 3

**Summary:**

The paper introduces Model-Based RL with Action Chunks (MAC), an offline RL algorithm designed to address two critical challenges: compounding errors in model-based rollouts and action selection that leads to out-of-distribution (OOD) model exploitation.

**Strengths:**

[S1] MAC's action-chunk model, combined with in-distribution rejection sampling, offers a promising solution to the problems of compounding model error (a key issue in MBRL) and OOD model exploitation (a key issue in offline RL). It has achieved excellent empirical performance in long-horizon and large-scale benchmark tests.

**Weaknesses:**

[W1] The main problem lies in the scalability of sampling with increasing action chunk length $n$. As $n$ increases, the dimensionality of the action chunk space also increases, making it exponentially more difficult to find high-value, in-distribution action sequences by rejection sampling. This sampling challenge may weaken the "return maximization" aspect of RL and may cause the agent's final behavior to be closer to behavior cloning (BC) rather than RL.

**Questions:**

[Q1] Following Weakness 1: How does the simpler reward- or goal-conditioned BC policy trained on the same flow-based architecture compare to the proposed rejection sampling plus value function approach? This comparison will help reveal the true advantage of value-based return maximization over the BC baseline policy.

[Q2] Given that performance peaks at an intermediate chunk length ($n=10$) and degrades with larger sizes ($n=25$), could the method be improved on large $n$ by using a mixture of chunk sizes, similar to the idea presented in [(1)](https://arxiv.org/pdf/2412.11253)? This might allow the agent to balance the low compounding error of large chunks with the more effective sampling of smaller chunks.

---

> ### Author Response · Authors · 2025-11-25
> **Response to Reviewer 9Rxo**
>
> Thank you for your constructive feedback. We address your questions and concerns in detail below and have revised the paper accordingly (please refer to the updated PDF). Specifically, we believe the added baseline (FBC) and clarifications have strengthened the paper. Please let us know if you have any additional concerns or questions. If we have fully addressed your concerns, we would appreciate it if you could update the score accordingly.
>
> &nbsp;
>
> **Q1. How does the simpler reward- or goal-conditioned BC policy trained on the same flow-based architecture compare to the proposed rejection sampling plus value function approach?**
>
> To answer the question, we provide the comparison with the Flow-based Behavior Cloning (FBC) baseline introduced in [1], which trains a BC policy with the same flow-based architecture without value learning, in the table below. MAC outperforms FBC across all long-horizon manipulation tasks, confirming that the value-guided rejection sampling provides clear performance gains over pure BC.
>
> | task | MAC | FBC |
> |---|---|---|
> | cube-octuple-play-oraclerep-v0 | 30.2 | 0.0 |
> | puzzle-4x5-play-oraclerep-v0 | 98.6 | 0.4 |
> | humanoidmaze-giant-navigate-oraclerep-v0 | 0.0 | 6.6 |
> | | | |
>
> &nbsp;
>
> **Q2. Could the method be improved on large $n$ by using a mixture of chunk sizes, similar to the idea presented in (1)? This might allow the agent to balance the low compounding error of large chunks with the more effective sampling of smaller chunks.**
>
> We agree that mixing chunk lengths could balance compounding error reduction with easier policy learning. Our design goal, however, was to keep hyperparameters minimal and consistent across tasks. We appreciate your suggestion, and believe that it will be great to explore adaptive chunking strategies to further enhance scalability in future works.
>
> &nbsp;
>
> **W1. The main problem lies in the scalability of sampling with increasing action chunk length. As $n$ increases, the dimensionality of the action chunk space also increases, making it exponentially more difficult to find high-value, in-distribution action sequences by rejection sampling. This sampling challenge may weaken the "return maximization" aspect of RL and may cause the agent's final behavior to be closer to behavior cloning (BC) rather than RL.**
>
> Thank you for pointing out the potential limitations of our work. Rejection sampling mitigates the problem of finding the high-value actions, because the search space is restricted to in-distribution action sequences, rather than having an exponential search space over all possible actions (as policy gradient methods). Still, we agree that rejection sampling and value learning may potentially become challenging for very large chunk sizes (in practice, we found that moderate chunk sizes (namely, 10) offer a good trade-off in our experiments). We added the discussion of this point in Section 6 (L494).
>
> &nbsp;
>
> **References**
>
> [1] Seohong Park et al., Horizon Reduction Makes RL Scalable, NeurIPS 2025

---

### Author Response · Authors · 2025-11-25
**Response to all reviewers**

We thank all reviewers for their constructive feedback! We included new experimental results, such as results for D4RL, more ablation experiments, and additional baselines. Moreover, we provide a clearer discussion of the method’s novelty, and its complexity. All updates in the revised PDF are marked in red.

Here, we summarize major updates and clarifications.

* D4RL results (Table 14, Section C)
* Training curves (Figure 5-7, Section D)
* Ablation experiment on using $\pi_\theta$ (Table 15, Section E)
* Experiment on the correlation between model error and the performance (Table 16, Section E)
* Comparison with ADMPO (Response for Reviewer P1Zz, results will be uploaded to the paper as the results for the full benchmark is available)
* Wall-clock time for training and inference of MAC and prior MBRL methods (Table 4-5, Section A.2)
* Additional discussion on the limitations of MAC (Section 6)
* Clarification on the novelty of MAC (Response for Reviewer ZGYu, P1Zz)

---

### Meta-Review · Area_Chair_MyJM · 2025-12-29

**Summary:**

The paper proposes a new offline model-based RL algorithm that enables accurate long roll-outs. The algorithm has two key ingredients: action chunking to mitigate error accumulation, and rejection sampling from behavior policy to avoid out-of-distribution errors.

**Reviewer Concerns:**

There is a general concern that action chunking has been a hot topic recently and works with similar ideas exist. Other technical concerns include questions on efficiency (e.g., whether policy optimization needs to search over an exponential space), algorithmic complexity (that the method needs to train many networks), and the lack of results on D4RL (added in revision).

One problem that no reviewer seems to touch on is that the work assumes an "expressive behavior policy". This is somewhat strong and the paper's framing around this issue introduces confusion: the behavior policy is given as part of the problem definition in offline RL (and sometimes deterministic), and not something we can change. Perhaps the authors meant to say that the architecture of the distilled policy is expressive, which is a totally different thing. In fact, even if the true behavior policy is deterministic, the distilled policy can be legitimately stochastic if the original policy takes different actions in similar states; the OPE community has some discussion around this issue which the authors can check out, see e.g., Importance Sampling Policy Evaluation with an Estimated Behavior Policy by Hanna et al.

**Reviewer Scores:**

The scores are 8(updated from 4)/6/6/4

---

### Decision · Program_Chairs · 2026-01-26

Accept (Poster)